# QUESTBENCH: CAN LLMS ASK THE RIGHT QUESTION TO ACQUIRE INFORMATION IN REASONING TASKS?

## ABSTRACT

Large language models (LLMs) have mastered a wide range of reasoning tasks, with an underlying assumption that the tasks are well-specified for LLMs to reach solutions. In reality, queries and instructions to LLMs often contain *incomplete* or *underspecified* information. Therefore, LLMs need to be able to actively acquire missing information, e.g., by asking clarifying questions, ideally seeking the minimally sufficient piece of information. To assess whether LLMs possess this ability, we construct QUESTBENCH, a set of underspecified reasoning tasks that can be solved by asking at most a single question. We frame the tasks as constraint satisfaction problems with missing variable assignments, where the exact model response cannot be determined unless certain variables' values are acquired. This framework specifically targets tasks where uncertainty stems from missing information, rather than semantic ambiguity in language. QUESTBENCH includes (1) Logic-Q: Logical reasoning tasks where one proposition is missing, (2) Planning-Q: PDDL planning problems where the initial state is partially observed, and (3) GSM-Q: Grade school math problems where one variable assignment is missing. Each task presents multiple choices of possible questions, only one of which is correct. We evaluate Gemini and GPT-4o models and find that they achieve 20 – 30% accuracy in both zero-shot and few-shot settings. When evaluating GPT-4-o1 on a subset of our data, we find that it is only 41 – 44% accurate, despite using state-of-the-art inference-time reasoning techniques. When investigating characteristics of QuestBench, we find that LLMs struggle with tasks that are computationally expensive for traditional search-based CSP solvers. Our analyses reveal a negative correlation between LLM accuracy and solver runtime complexity, suggesting that LLMs may share similar limitations to CSP solvers.

## 1 INTRODUCTION

Large language models (LLMs) have become popular tools for solving reasoning tasks such as math (Cobbe et al., 2021; Hendrycks et al., 2021; Li et al., 2024a), logic (Zhang et al., 2023; Chen et al., 2024b; Creswell et al., 2022) and planning/coding (Curtis et al., 2024a; Silver et al., 2024; Wang et al., 2024a; Austin et al., 2021; Chen et al., 2021). An underlying assumption for these tasks is that they are well-specified, i.e., all premises are given such that there exist answers.

Yet this assumption does not always hold in practice: users can miss conditions when asking math questions; robots usually do not observe the entire initial state. In these circumstances, proactively obtaining relevant information, e.g., by asking questions, is an important skill that LLMs need to master.

Do LLMs have the ability to identify what information they should gather? Existing benchmarks mostly focus on

> Aleena subscribed to a streaming service that charges her $140 per month. If the streaming company charged her the initial amount for the first three months and then charged her 10% less money on the remaining half of the year, calculate the total amount she had paid for the streaming service by the end of the year.
> ---
> What information must be gathered?
> (A) The amount the streaming company charged from June to December.
> (B) The amount the streaming company charged in July.
> (C) The amount the streaming company charged from July to December.
> (D) The amount the streaming company charged in June.

Figure 1: An example of tasks in QUESTBENCH.

resolving ambiguity in language (Kuhn et al., 2023a) and disambiguating intents in task oriented-dialogues (Rastogi et al., 2020; Budzianowski et al., 2018). Some recent work has looked at asking

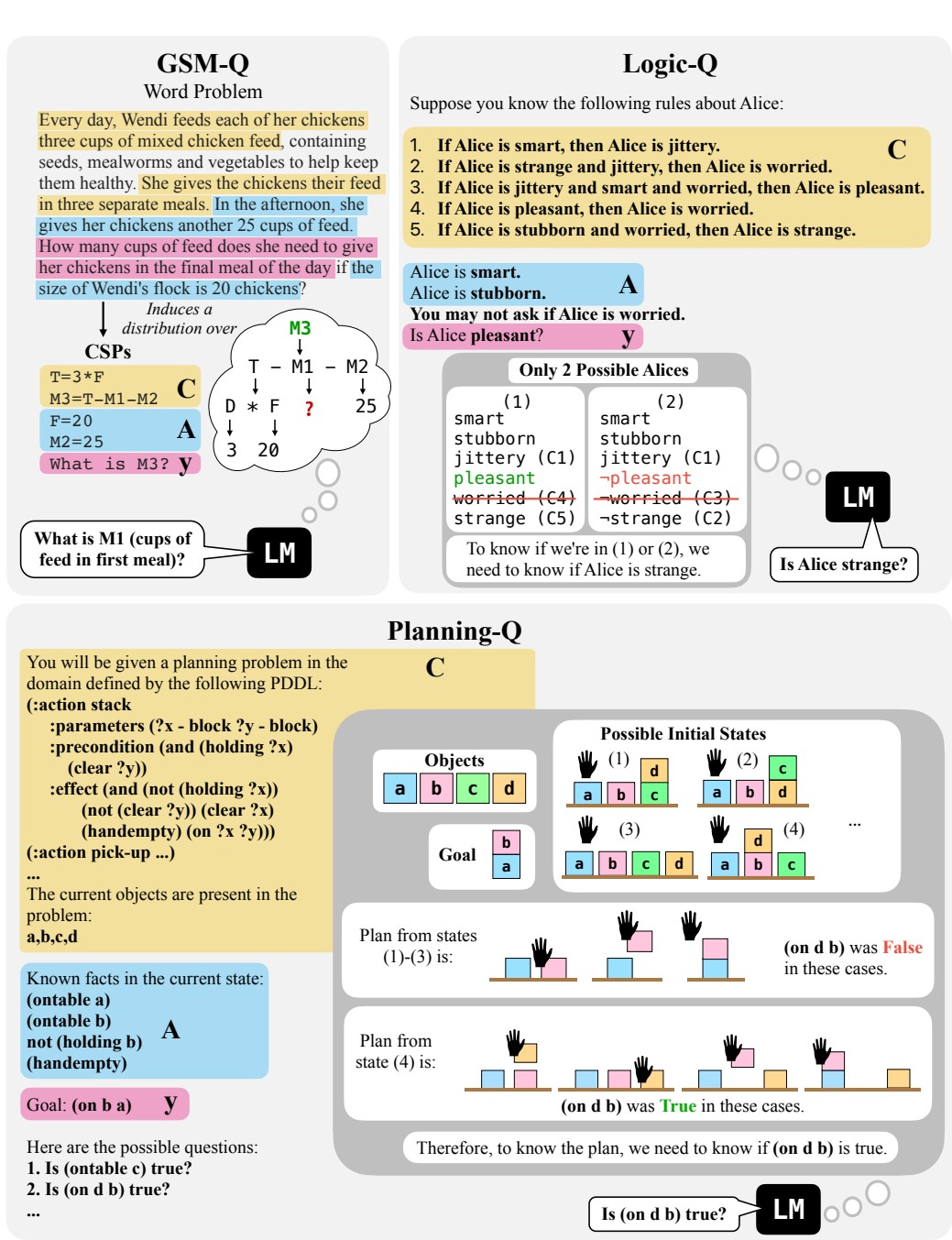

Figure 2: A sample of underspecified problems from three domains in QUESTBENCH. We construct three types of underspecified reasoning tasks, where each task can be represented as a constraint satisfaction problem (CSP) and exactly one missing piece of information is required to solve the task. At the top left, we show an example from GSM-Q, at the top-right, an example from Logic-Q, and at the bottom, an example from Planning-Q. $C$ represents constraints of the CSP, $A$ represents assignments to a subset of the variables, and $y$ is the target variable.

questions for disambiguating tasks (Handa et al., 2024; Li et al., 2023), factual questions (Min et al., 2020) and preferences (Chen et al., 2024a). In these cases, the exact information necessary to acquire may depend on individuals and populations (Aroyo & Welty, 2015; Davani et al., 2022; Basile et al., 2021; Sandri et al., 2023; Wan et al., 2023). E.g., a person asks for dinner recommendation, and the LLM asks "how about tuna nigiri?", which might be helpful for someone in Japan, but not for someone who is vegan. That means, there is often no ground truth for the question that must be asked. These factors make it challenging to use such existing benchmarks to reliably assess the ability of LLMs to identify what information is required, especially for complex reasoning problems.

To address this gap, we construct QUESTBENCH: a collection of question asking benchmarks based on *underspecified constraint satisfaction problems*. QUESTBENCH is a multi-choice dataset illustrated in Figure 1, and it aims to reliably evaluate how well LLMs can reason about missing information and proactively ask the right clarification questions. The benchmarks include

• Logic-Q: logical reasoning tasks where a missing proposition's truth value is needed to determine whether a claim is true or false,
• Planning-Q: PDDL planning problems with partially observed initial states, where additional observations are needed to disambiguate a path to a goal state, and
• GSM-Q: grade school math problems that are missing a condition for deriving the solution.

Each problem in QUESTBENCH possesses an *objective* single piece of missing information[1], making it ideal for quantitative evaluation. QUESTBENCH problems exhibit varying levels of difficulty, determined by the search depth and width of their corresponding constraint satisfaction problems (CSPs). Examples can be found in Figure 2. This ensures a thorough understanding of the aspects in which the models excel and those that require improvement.

We evaluate the performance of Gemini Pro 1.5 (Gemini Team Google, 2024), GPT-4o (OpenAI, 2023) and GPT-4-o1 on QUESTBENCH. These models are the current state-of-the-art models judged by Chatbot Arena (Chiang et al., 2024). These models achieve between 15–44% and 9–41% for Logic-Q and Planning-Q respectively, showing that there is significant room for improvement.

Our contributions include (1) a constraint satisfaction framework enabling focused evaluation of underspecification in LLMs, (2) a benchmark QUESTBENCH for evaluating the information gathering abilities of LLMs, (3) empirical evaluation and analyses of LLM performance on QUESTBENCH. The dataset can be found at `anonymous`.

## 2 BACKGROUND AND RELATED WORKS

**Ambiguity in User Requests**   Natural language queries posed by human users can be ambiguous for many different reasons. Prior work has examined ambiguity in the context of semantic uncertainty (Kuhn et al., 2023b), factual question-answering (Min et al., 2020), intents in task-oriented dialogues (Rastogi et al., 2020; Budzianowski et al., 2018; Zhang et al., 2024b), personalized human preferences and tasks (Li et al., 2023; Handa et al., 2024; Chen et al., 2024a). Zhang et al. (2024a) introduces a taxonomy of ambiguity including categories like unfamiliarity (lack of knowledge), semantics and possible question types such as who, when, what.

In this paper, we focus on *underspecification*, where the user has not provided enough information for the LM to fulfill the request. This may be because the user is unaware of what information is necessary to complete a task, or is unaware of what information the LM does not know.

**Importance of Information Gathering**   For humans, Chouinard et al. (2007) finds that actively gathering information (e.g., by asking questions) is an indispensable skill for solving problems under uncertainty, and may even play a crucial role in cognitive development of children.

For autonomous systems, information gathering abilities are especially crucial since the system can often be uncertain. In the literature of AI/ML, researchers have extensively explored strategies to sequentially acquire data for tasks like concept learning (Sammut & Banerji, 1986; Angluin, 1988), active learning (Cohn et al., 1996; Settles, 2009; Houlsby et al., 2011; Gal et al., 2017; Ren et al., 2021), Bayesian optimization (Kushner, 1964; Močkus, 1974; Auer, 2002; Srinivas et al., 2010;

---

[1]Though in practice more information may be missing, evaluations with one piece of missing information serve as an upper bound on performance for queries with multiple pieces of missing information.

Hennig & Schuler, 2012; Wang & Jegelka, 2017; Garnett, 2023; Wang et al., 2024b), reinforcement learning (Kaelbling et al., 1996; Ghavamzadeh et al., 2015; Sutton, 2018), and robot planning with partially observable states (Kaelbling et al., 1998; Kaelbling & Lozano-Pérez, 2013; Phiquepal & Toussaint, 2019; Curtis et al., 2024b). Wu (2023) argues AI assistants should also ask questions in the presence of uncertainty, specifically in the context of generating code.

Information gathering is also an important skill for LLMs as they often operate under uncertainty. Our work proposes basic reasoning tasks to evaluate how well LLMs master such skills.

**Question-asking Problems and Techniques for LMs**   Prior works have introduced question-asking methods for subjective or knowledge-based tasks with ambiguous user requests, such as "What is a good pasta recipe" (Andukuri et al., 2024) or "Who won the US open?" (Zhang & Choi, 2023; Pang et al., 2024). Li et al. (2023); Kuhn et al. (2023a) directly prompt a language model to ask questions. Piriyakulkij et al. (2024) augments this process by sampling questions with the highest information gain, while Andukuri et al. (2024) encourages LMs to produce useful questions through self-training. Grand et al. (2024) parses questions into symbolic programs before computing information gain on top of programs. Handa et al. (2024) computes information gain with respect to preferences represented as Bayesian linear models. Hu et al. (2024) introduces a method based on simulating entire conversation trajectories, computing information gain at each turn, and propagating rewards through a conversation. Zhang & Choi (2023) decomposes question-asking into three stages: (1) determining when clarification is needed, (2) determining what clarifying question to ask, and (3) responding accurately with the new information gathered through clarification. Zhang et al. (2024b) trains agents to ask for clarification in travel planning settings.

To our best knowledge, these methods either do not apply to our tasks or require significant modifications (such as simulating users) to be applied to the underspecified reasoning tasks in our benchmark. We are not aware of existing methods that solve underspecified CSPs defined in our work.

## 3   PROBLEM FORMULATION

Consider the following user request:

**Example 3.1.** *Please solve the math problem: Janet had a few eggs (variable $x_0$) and ate one (variable $x_1$). How many eggs does she have now (target variable $y$)?*

While it is clear that the arithmetic word problem can be parsed to $y = x_0 - x_1, x_1 = 1$, the LM cannot compute target variable $y$ without knowing the value of variable $x_0$. Other examples can be found in Figure 2. In these cases, the desired behavior is for the LM to ask the minimal set of questions that enables it to respond to the user query.

### 3.1   SEMANTIC AMBIGUITY VS. UNDERSPECIFICATION

In this paper, we focus on problems that are semantically equivalent to constraint satisfaction problems,[2] which allow us to formalize underspecification in a well-defined and rigorous manner.

Solving a problem thus comprises of two stages:
1. A natural language query is *parsed* into a set of variables and constraints, and a *target variable* corresponding to the desired response. For example, in Example 3.1, there are variables $x_0$ (initial eggs), $x_1$ (eaten eggs), target variable $y$ (current eggs), and a constraint $y = x_0 - x_1$.
2. The model solves for the target variable in the CSP.

The model might not know the answer for the target variable due to two possible reasons.
1. Semantic ambiguity: There are multiple semantic interpretations of the problem at the parsing stage, inducing a distribution over possible CSPs.
2. Underspecification: For a given CSP, some variable assignments or constraints may be missing that makes it impossible to solve for the target variable.

Semantic ambiguity has been treated extensively in prior work (Kuhn et al., 2023b), and may vary based on different human's interpretations due to reasons including mental states or demographics.

---

[2]Note: open-ended queries can be viewed as a combination of *soft* constraints.

This work only considers *underspecification*, which can be evaluated by formulating problems as CSPs with missing information.

## 3.2 Formalizing Underspecification

We formalize underspecification and the information gathering objective below. This formalism is critical for constructing tasks to evaluate the question asking skills of models. First, we define a CSP as a tuple $\langle X, D, C, A, y \rangle$ with:

- $X = \{x_i\}_{i=1}^N$ is a set of $N$ variables.
- $D = \{D_i\}_{i=1}^N$ is a set of domains for each variable in $X$.
- $C = \{c_j\}_{j=1}^M$ is a set of $M$ constraints. Each constraint is associated with a list of variables in $X$ and specifies the values that can be assigned to those variables.
- $A = \wedge_{i \in I_A}(x_i = a_i)$ is a proposition stating that for each $i \in I_A \subseteq [N]$, variable $x_i$ is assigned a value $a_i \in D_i$.
- $y$ is a target that the user requests to be solved.

In Example 3.1, the variables are $\{x_0, x_1, y\}$ and $y$ is also the target. The constraint $c_0$ is $y = x_0 - x_1$.

We use the shorthand $c_j(\boldsymbol{x}_j)$ to denote $c_j([x_i]_{i \in I_j})$. Variable $y$ must satisfy $\bigwedge_{j \in [M]} c_j(\boldsymbol{x}_j) \wedge A$.

**The *Known* predicate.** For convenience, we introduce the *Known* predicate. *Known*$(x)$ means that the value of variable $x$ is known. For example, if variables are assigned values, their values are known. That means, $A \implies \wedge_{i \in I_A} Known(x_i)$. If the value of a variable can be derived from the CSP, its value is also known. In Example 3.1, once values to both variable $x_0$ and $x_1$ are assigned, the value of $y$ is uniquely determined since $y = x_0 - x_1$. We consider the value of $y$ known in this circumstance, i.e., *Known*$(y)$ = True.

Conversely, $\neg Known(x)$ means that the value of variable $x$ is unknown, i.e., its value is not assigned and cannot be derived from the CSP. For example, for the user query "$a = 1, y = a + b$. What is the value of $y$?", we have $\neg Known(y)$ = True since $\neg Known(b)$ = True.

We can then formally define underspecified CSPs as follows.

**Definition 3.1.** *A CSP is underspecified if and only if* $\bigwedge_{j \in [M]} c_j(\boldsymbol{x}_j) \wedge A \implies \neg Known(y)$.

We use *Known*$(\mathcal{X})$ over a set of variables $\mathcal{X}$ to denote that the values of all variables in $\mathcal{X}$ are known, i.e., *Known*$(\mathcal{X}) = \bigwedge_{x \in \mathcal{X}} Known(x)$.

Knowing the values for a set of variables can potentially be sufficient to determine the value of $y$, and we define such sets as follows.

**Definition 3.2.** *A set of variables $\mathcal{X} \subseteq X$ is a sufficient set if and only if*

$$\bigwedge_{j \in [M]} c_j(\boldsymbol{x}_j) \wedge A \wedge \text{Known}(\mathcal{X}) \implies \text{Known}(y).$$

The **objective** of efficient question asking for an underspecified CSP is to find a smallest sufficient set[3], i.e., $\hat{\mathcal{X}} = \arg\min_{\mathcal{X} \subseteq X} |\mathcal{X}|$ s.t. $\mathcal{X}$ is sufficient. That means, in order to solve for the target $y$, the model only needs to ask questions about the variables in a smallest sufficient set.

For example, in the context of $a = 1, y = a + b$, we have *Known*$(b) \implies Known(y)$, so the information to seek is the value of $b$. In a logic problem, every variable can take on True or False values. If we have $a \vee b \implies y, y \implies b$ and $\neg a$, we cannot determine the value of $y$. But if we know the value of $b$, not matter if $b$ is True or False, we know the value of $y$. That means, *Known*$(b) \implies Known(y)$.

Finally, we define the special case where the size of the smallest sufficient sets is 1.

**Definition 3.3.** *An underspecified CSP is a 1-sufficient CSP if the size of its smallest sufficient sets is 1. We call those smallest sufficient sets 1-sufficient sets.*

The problems in all datasets we construct in §4 are 1-sufficient CSPs.

---

[3]There could be many smallest sufficient sets for an underspecified CSP.

# 4 DATASET CONSTRUCTION

To examine LLMs capabilities at the easiest setting, we construct datasets where each task is a **1-sufficient CSP** as defined in Definition 3.3 (samples are shown in Figure 2). Note the 1-sufficient set need not be unique (there may be multiple size-1 smallest sufficient sets). During evaluation, we consider a LLM's behavior to be correct if they produce a variable in any 1-sufficient set.

We construct 1-sufficient CSP problems in three domains: SimpleLogic, Planning, and grade-school math (GSM), which lie at various points along the real-synthetic axis. SimpleLogic is a simple, fully synthetic setting to arbitrarily generate any logic problems. Planning is a more realistic domain where external physical commonsense knowledge may need to be leveraged. Finally, GSM is a realistic setting of concrete math problem. See Table 1 for dataset statistics.

| | # Total Tasks |
|---|---|
| Logic-Q | 1152 |
| Planning-Q | 7500 |
| GSM-Q | 570 |

Table 1: Dataset sizes.

## 4.1 LOGIC-Q

SimpleLogic (Zhang et al., 2023) is a propositional logic benchmark, which consists of

1. a set of *rules* about an imaginary person named Alice, for example: *If Alice is strange and jittery, then Alice is worried.*
2. a set of *properties* that hold true about Alice, for example: *Alice is strange. Alice is not jittery.*
3. a question about an unknown property of Alice, for example: *Is Alice worried?*

**Problem Definition.** We can define a CSP in this domain as follows:

- $X$ is a set of all the potential properties of Alice that appears in all rules (e.g. *strange, jittery, worried)*.
- $D = \{\{\text{TRUE}, \text{FALSE}\} \forall x \in X\}$. Each property in $X$ can be either be true or false.
- $A$ is the set of properties that we know to be true about Alice (e.g. *strange*, $\neg$*jittery*).
- $C$ is the set of rules about Alice. $C$ takes the form of a conjunction of implicative constraints (e.g. in the above example, *strange* $\wedge$ *jittery* $\implies$ *worried*).

$$C = \bigwedge_{i \in [M]} c_i = \bigwedge_{i \in [M]} \left( \left( \bigwedge_{j \in [M_i]} x_{i,j} \right) \implies x_{i,M_i+1} \right)$$

- $y$ is the property that we are being asked about, e.g. *worried* in the above example.

Because the original CSPs in the dataset were fully specified, we discard the $A$'s that are already present in the dataset. We then construct new $\tilde{A}$'s such that a single additional property of Alice is sufficient to fully determine whether the goal property is true or false, such that $\langle X, D, \tilde{A}, C, y \rangle$ form a 1-sufficient CSP. We detail the dataset construction procedure in Appendix A.1.

## 4.2 PLANNING-Q

We adapt the blocks world problem from PyperPlan (Alkhazraji et al., 2020), a pick-and-place task involving $n$ blocks which can be set on the table or stacked on top of each other.

We use the following predicates:

```
(ontable ?a), (clear ?a), (handempty), (holding ?a), (on ?a ?b).
```

Each predicate can be applied to any blocks to construct an atom, e.g., (ontable a). A state is a conjunction of atoms or their negations[4], e.g., for two blocks a and b, a state can be

```
(ontable a), (holding b), ¬(on a b).
```

At each state, there are a set of actions that can potentially be applied to it to transition to another state. Each action has a precondition specifying atoms the state must satisfy before applying the action, and an effect specifying atoms the next state must satisfy after applying the action. An example of an action:

```
stack(?a, ?b)
    :precondition (and (holding ?a) (clear ?b))
```

---

[4]Note that not all states are valid under this definition.

```
:effect (and (not (holding ?a)) (not (clear ?b))
             (clear ?a) (handempty) (on ?a ?b))
```

Given an initial state and a goal state, the model is expected to plan the *shortest* action sequence to the goal state, e.g., `[pick-up(b), stack(b,a)]`.

**Problem Definition**  The problem can be expressed as a CSP $\langle X, D, C, A, y \rangle$ as follows:[5]

- $X$ is the set of all atoms (predicates applied to objects) in the initial state.
- $D = \{\{\text{TRUE}, \text{FALSE}\} \, \forall x \in X\}$. Each atom must be either true or false.
- $A$ is the set of atoms with known values in the initial state.
- $C$ is the set of constraints. For any sequence of actions $[q_t]_{t \in [n]}$ and their corresponding sequence of states $[s_t]_{t \in [n]}$, each pair of current state $s_t$ and next state $s_{t+1}$ must satisfy the precondition and effect of action $q_t$, i.e.,

$$\forall p \in \text{PRE}(q_t), \, s_t \implies p \qquad \text{[[preconditions of } q_t \text{ must hold in } s_t\text{]]}$$
$$\forall e \in \text{EFFECTS}(q_t), \, s_{t+1} \implies e \qquad \text{[[effects of } q_t \text{ must hold in } s_{t+1}\text{]]}$$

- $y$ is the shortest action sequence from the initial state to the goal state.

We construct problems such that the initial state is potentially *underspecified* – we only know the values of certain atoms of the initial state, and must ask the value of at most a single other atom in order to disambiguate the shortest action sequence.

Some problems are *fully specified CSPs* – i.e., the shortest action sequence can be fully determined regardless of uncertainty about the initial state. For example, even if we do not know whether block `b` is on top of another block or on the table, that information does not affect what the shortest action sequence will be. In these cases, we expect the LLM to output "*No questions needed.*"

Some problems are *1-sufficient CSPs* – the LLM must ask for the value of an atom, which will be enough to fully determine the shortest action sequence, regardless of the value of the atom.

Details about dataset construction can be found in Appendix A.2.

### 4.3   GSM-Q

In general, grade-school math problems can be parsed into simple algebra problems where a sequence of variable substitutions can fully solve the problem. We construct underspecified grade-school math problems from GSM-Plus' "distractor" setting (Li et al., 2024a), which was derived from adding a single distractor sentence to math problems in GSM8k (Cobbe et al., 2021) that is irrelevant to deriving the goal. This allows us to isolate whether LLMs can identify which specific variables are relevant to a goal.

**Problem Definition.**  Given a generic GSM8k word problem, we can parse it into a 1-sufficient CSP $\langle X, D, C, A, y \rangle$ where

- $X$ is a set of variables in the problem.
- $D$ is possible values $x$ can take on, generally $\mathbb{N}, \forall x \in X$.
- $C$ is a set of equations relating variables in the problem to each other.
- $A = \tilde{A}$ variable assignments from the problem with a single variable withheld.
- $y$ variable the user wants to solve for.

An example can be seen in Figure 2 (left).

**Construction.**  Unlike the datasets introduced so far, we use human annotators to construct this dataset. We construct 1-sufficient CSPs as follows: First, we ask annotators to try and solve the word problems on their own. If they cannot solve the problem due to semantic ambiguity, or answer

---

[5]Note there were alternative ways of constructing 1-sufficient planning CSPs that we considered: first, we considered asking LLMs to discover what must be known about the initial state such that there is *an* action sequence (not just the shortest one) to the goal. However, in BlocksWorld all initial states can get to the goal state through some path, making this a trivial task. Second, we also considered *giving* the trajectory to the LLM and asking it to determine whether the trajectory was the shortest path for an underspecified initial state. However, this made the problem much easier as the LLM simply had to follow an existing plan, rather than performing search to discover distinct plans.

| | | Logic-Q | | Planning-Q | | GSM-Q |
|---|---|---|---|---|---|---|
| | | Full | Subset | Full | Subset | Full |
| ZS | o1-preview | - | 44% | - | 41% (44%) | - |
| ZS | GPT-4o | 27.39% | 24% | 14.61% (14.55%) | 9% (12%) | 84.74% |
| | Gemini Pro | 29.13% | 23% | 19.80% (20.36%) | 21% (24%) | 81.23% |
| | Gemini Flash | 16.78% | 11% | 8.52% (11.76%) | 9% (16%) | 45.26% |
| ZS + CoT | GPT-4o | 28.04% | 28% | 10.49% | 8% | 93.51% |
| | Gemini Pro | 29.13% | 19% | 21.27% | 28% | 94.91% |
| | Gemini Flash | 18.70% | 11% | 9.33% | 5% | 90.88% |
| 4S | GPT-4o | 25.57% | 22% | 11.48% | 9% | 86.49% |
| | Gemini Pro | 26.35% | 22% | 18.80% | 21% | 58.13% |
| | Gemini Flash | 15.48% | 18% | 18.29% | 18.18% | 60.00% |

Table 2: Language model accuracies at predicting the right question in QUESTBENCH. For the Planning-Q domain, we additionally evaluate a setting where additional physical constraints are included in the prompt in the 0-shot (ZS) setting, the results of which appears in parentheses. We include results on the full dataset, as well as results on a smaller, curated subset of Logic-Q and Planning-Q, which we use to evaluate GPT4-o1-preview. While o1 substantially outperforms other models, it still struggles to perform beyond 50%, indicating large room for improvement.

the problem in a way that doesn't match the original answer in GSM-Plus (due to interpreting the problem differently, or erroneous problems in GSM-Plus), we discard the problem entirely.

Next, we ask annotators to try and parse each math word problem into a set of variables $X$, constraints $C$, assignments $A$, and a goal variable $y$. We assume domains $D$ for each variable is the set of natural numbers. We had three different annotators provide CSPs for each math problem, as different annotators may have different interpretations of a problem, resulting in different CSPs. We further perform automated checks to ensure the annotated CSPs actually result in the correct answer, discarding any CSP which we weren't able to parse or provided incorrect answers. We use all valid CSPs resulting from this process, including different CSPs corresponding to the same math problem. This gives us the set of fully specified CSPs.

To make these CSPs *underspecified*, we withhold both *distractor* variable assignments $\{d_i\}_{i=0}^n$ that *aren't* essential to computing $y$, as well as a single variable assignment $a \in A$ that *is* required for computing the goal, creating $\tilde{A} = A \setminus (\{d_i\}_{i=0}^n \cup \{a\})$. Asking about the value of the variable corresponding to $a$ is necessary and sufficient for deriving the value of target variable $y$.

We directly condition on the underspecified CSPs in order isolate how well LLMs are at reasoning about underspecification, without the presence of semantic ambiguity.

**Annotation Details**   We recruited a total of 21 annotators (11 male, 10 female) from five countries to annotate our tasks. Annotators were all fluent in English and between the ages of $25 - 45$. We paid annotators an average of $27 - $55 per hour. Details including full instructions provided to annotators and screenshots of the annotation interface for each task, can be found in Appendix A.3.

## 5   RESULTS

We can restrict LLMs to ask a finite set of questions (about the value of each variable in $X$), and expect them to ask a question about a variable in the sufficient set of $A$. For the Planning-Q domain, we additionally have the option to include physical constraints (e.g. (ON A B)$\rightarrow \neg$(ON B A)), which are inferrable from the pre- and post-conditions of each action written in the PDDL, but aren't explicitly stated) in the prompt.[6]

We evaluate GPT-4o, Gemini 1.5 Pro, Gemini 1.5 Flash-S, in zero-shot (ZS), chain-of-thought (CoT), and four-shot settings (4S), the results of which are shown in Table 2.

---

[6]We infer that some of these constraints were probably learned as common-sense knowledge during LLM pre-training. For those that weren't learned through common-sense, this is equivalent to providing LLMs with shortcuts in reasoning.

| | | Logic-Q | | | | Planning-Q | | | | GSM-Q | | | |
|---|---|---|---|---|---|---|---|---|---|---|---|---|---|
| | | $d$ | $|X|$ | $|C|$ | $\mathbb{E}_{BF}$ | $d$ | $|X|$ | $b$ | $p_g$ | $d$ | $|X|$ | $|C|$ | $p_g$ |
| | o1-Preview | **-0.41** | **-0.47** | **-0.44** | **-0.43** | **0.22** | 0.15 | 0.15 | **0.57** | - | - | - | - |
| ZS | GPT-4o | **-0.19** | **-0.16** | **-0.11** | **-0.26** | -0.00 | **-0.07** | **-0.07** | **0.45** | **-0.14** | **-0.18** | **-0.29** | **0.18** |
| | Gemini Pro | **-0.24** | **-0.19** | **-0.12** | **-0.35** | 0.01 | **-0.11** | **-0.11** | **0.30** | -0.07 | **-0.15** | **-0.26** | **0.15** |
| | Gemini Flash | **-0.22** | **-0.26** | **-0.19** | **-0.32** | **-0.12** | **-0.13** | **-0.13** | **0.21** | **-0.09** | **-0.49** | **-0.37** | **0.49** |
| ZS + CoT | GPT-4o | **-0.20** | **-0.18** | **-0.15** | **-0.24** | **-0.07** | **-0.10** | **-0.10** | **0.40** | 0.07 | 0.06 | **-0.15** | -0.06 |
| | Gemini Pro | **-0.24** | **-0.21** | **-0.13** | **-0.37** | -0.03 | **-0.04** | **-0.04** | **0.16** | 0.07 | 0.07 | **-0.11** | -0.07 |
| | Gemini Flash | **-0.22** | **-0.25** | **-0.14** | **-0.34** | **-0.08** | **-0.09** | **-0.09** | **0.31** | 0.02 | -0.01 | **-0.10** | 0.01 |
| 4S | GPT-4o | **-0.20** | **-0.18** | **-0.20** | **-0.30** | **-0.06** | **-0.10** | **-0.10** | **0.41** | **-0.09** | **-0.33** | **-0.34** | **0.33** |
| | Gemini Pro | **-0.15** | **-0.11** | -0.03 | **-0.31** | **0.06** | **-0.03** | **-0.03** | **0.51** | **-0.09** | **-0.40** | **-0.45** | **0.40** |
| | Gemini Flash | **-0.15** | **-0.08** | 0.04 | **-0.20** | 0.01 | **-0.12** | **-0.12** | **0.33** | -0.05 | **-0.51** | **-0.36** | **0.51** |

Table 3: Spearman's rank correlation coefficient between various axes and accuracy for at predicting the right question. For the Planning-Q domain, we additionally evaluate a setting where additional physical constraints are included in the prompt. **Bolded** values indicates a statistical significance correlation between the axis and model accuracy ($p < 0.05$).

Generally, we find that even the largest and most recent models (GPT-4o, Gemini 1.5 Pro) struggle to perform beyond 30% on our Logic-Q and Planning-Q domains. It is not always the case that few-shot and Chain of thought (CoT) can improve the performance. Furthermore, we found that the best models (GPT4-o and Gemini-Pro) were fairly good at inferring missing variables in GSM-Q. Even Gemini Flash is able to achieve 90% on GSM-Q when augmented with chain-of-thought. We suspect that because of how GSM-Q was annotated, the task simply boils down to identifying unassigned variables missing from the left-hand-sides of any constraint (see Section 6).[7] Furthermore, the problems in GSM-Q generally have a small number of variables and constraints compared to problems in the other two domains, and thus appear to be sufficiently easy for SoTA models.

Additionally, we evaluate GPT4-o1 on our dataset. Due to the computational and financial constraints, we evaluate GPT-o1 on only 100 samples from each of Logic-Q and Planning-Q. We make sure to evenly sample from both easy and difficult problems when creating these 100-sample subsets. The results of each model on this subset is reported in the *subset* column of Table 2. We find that despite being optimized for reasoning, GPT4-o1 is only able to achieve ∼40% on this subset, while using 200-300 times more inference tokens. This indicates that even the most current in-context reasoning techniques have significant room for improvement on QUESTBENCH.

## 6 CORRELATION BETWEEN SEARCH COMPLEXITY AND LLM ACCURACY

The 1-sufficient CSPs can be solved through **brute-force (forward) search** or **backwards search** in general. Are LLMs using a similar mechanism or something completely different? In this section, we conduct preliminary investigation by estimating how much model's accuracy correlate with the human-understandable problem parameters (E.g., depth of the search).

We can approximately quantify the difficulty of each problem in these domains based on the runtime complexity of each algorithm. If there is a correlation between the factors that determine algorithmic complexity, and the performance of the LLM, this serves as a high-level signal for the types of mechanisms LLMs may be using to succeed in these tasks.

Specifically, we analyze the correlation between LLM performance and the following factors:

---

[7]We suspect the large jump from 45% to 90% when chain-of-thought is applied to Gemini Flash in GSM-Q is due to chain-of-thought enabling the LM to iterate through all the variables and check their absence.

| | Logic-Q | Planning-Q |
|---|---|---|
| Brute-force Search | $O((|X| + |C|)\mathbb{E}_{\text{BF}}/\delta)$ with probability $\geq \delta$ | $O\left(4^{|X|}b^2\right)$ |
| Backwards Search | $O\left(|X|^{|C|d}\right)$ | $O(b^{2d})$ or $O\left(3^{|X|d}\right)$ |

Table 4: Runtime complexities of brute-force and backwards search in Logic-Q and Planning held-out. Derivation details can be found in Appendices C.1 and C.2.

- $d$: depth of backwards search. In planning, $d$ also represents the maximum length of the shortest path from any possible initial state to the goal conditions.
- $|X|$: number of variables.
- $|C|$: number of constraints.
- $b$: number of blocks (for Planning-Q).
- $\mathbb{E}_{\text{BF}}$: expected number of guesses for brute-force search (determined by number of total variables $|X|$ divided by the number of sufficient sets).
- $p_g$: probability that a random question is correct, computed as the percentage of correct answers divided by the total number of possible questions.

We report Spearman's rank correlation coefficients between accuracy and these factors in Table 3. The full set of plots of accuracy across each factor can be found in Appendix D. We describe these factors in more detail below, including how they contribute to the runtime complexity of each algorithm in each domain below.

**Logic-Q** The complexities of Brute-force and Backwards search can be found in Table 4. Details of how these algorithms are implemented can be found in Appendix C.1.

Looking at Table 3, we find that performance is negatively correlated with backwards search depth, number of variables, number of constraints, and expected number of brute-force guesses. These correlations are statistically significant for most LLMs in this domain, indicating that in Logic-Q, brute-force and backwards search may serve as good approximations for the type of reasoning LLM are required to perform.

**Planning-Q** The complexities of Brute-force and Backwards search can be found in Table 4. Details of how these algorithms are implemented can be found in Appendix C.2.

Looking at Table 3, we find that performance is generally only slightly negatively correlated with $d$, $|X|$, $b$, and positively correlated for GPT4-o1-preview. This indicates that LLMs may be using mechanisms other than the backwards search or brute-force search in planning. This is reasonable as these search algorithms generally require exponential-time search, while LLMs cannot perform non-polynomial-time-computation within a polynomial amount of chain-of-thought (Li et al., 2024b).

**GSM-Q** We anticipate that our GSM-Q task is sufficiently easy without search. When examining the annotated CSPs, we find that only 25% of the problems were annotated to include extraneous irrelevant variables. Thus, most problems in GSM boil down to missing-variable detection: the model simply needs to detect which variable is missing from the left-hand-side of assignments and constraints, and ask about the value of that variable. Thus, of the metrics reported here, only $|C|$ is consistently correlated with task difficulty, indicating difficulty of iterating through the constraints.

## 7 DISCUSSION AND CONCLUSION

We introduce a collection of benchmarks for identifying underspecification in problems, which we posit is an essential skill necessary for LLMs' to perform tasks under uncertainty. We found that SOTA LLMs are relatively good at identifying missing information in simple algebra problems, but struggle with more complex tasks involving logic and planning. Their performance tends to decline with the increase of the size of solution spaces and search depths, if these problems were to be solved by a search algorithm. We conjecture that LLMs may possess search skills similar to breadth-first search or brute-force approaches, which become less effective as the search space expands. Future work includes understanding the mechanisms of LLMs to determine what information is missing in a given task, and how to enhance their ability to effectively seek that information. One possible method for higher performance is to use LLMs to extract the symbolic CSP for an underspecified task and then run search algorithms to find the right variable to clarify.

ETHICS STATEMENT

We foresee limited ethical risks with our benchmark, as it focuses on low-risk domains such as logical reasoning, blocks-world planning, and math problems. This work will enable progress towards LLMs that can work with humans to ask clarifying questions. We note when or if these techniques are eventually deployed into the real world, they can *alleviate* unforeseen risks through confirming with users before executing on a task. Furthermore, LLMs that can ask clarifying questions are likely more easily customizable and personalizable. However, this may also introduce risks if malicious actors can more easily use LLMs for malicious purposes; care must be taken to ensure these risks do not come to fruition, for example, by preventing LLMs from ever displaying certain behavior.

REPRODUCIBILITY

We plan to release the code and benchmark upon publication. All data construction details and annotation instructions can be found in the appendix. All models were evaluated with temperature 0 settings. GPT-4o and GPT-4-o1 models were queried through the OpenAI API.

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

**Limitations.** We construct simplified tasks with only one missing piece of information to precisely evaluate whether LLMs is able to identify that missing piece. Real-world user queries are much more complex. They often involve multiple missing pieces of information and inherent ambiguities of natural language. To address this, our work could be extended by modeling distributions over constraints on model responses. However, determining such distributions would likely require extensive user studies and experiments, which is beyond the scope of this work.

# A   DATASET CONSTRUCTION DETAILS

## A.1   LOGIC-Q

We create 1-sufficient CSPs out of SimpleLogic problems by first discarding the $A$'s that are already present in the dataset. We then identify all assignments $A^{(y)} = \{A_i^{(y)} : A_i^{(y)} \implies y\}_{i=0}^{M}$ to (a subset of) variables in $X\backslash y$ which would imply $y$ is true, and similarly all assignments $A^{(\neg y)} = \{A_i^{(\neg y)} : A_i^{(\neg y)} \implies \neg y\}_{i=0}^{M'}$ that imply $y$ is false. These sets are found through recursive backwards search starting from $y$ or $\neg y$, see Appendix A.1.1.

Once we have the full set of assignments which imply $y$, to make them underspecified, we examine all pairs of assignments $(A_i^{(y)}, A_j^{(\neg y)}) \in \{A_i^{(y)}\}_{i=0}^{M} \times \{A_i^{(\neg y)}\}_{i=0}^{M'}$ where $A_i^{(y)}$ and $A_j^{(\neg y)}$ differ on an assignment to a single variable $x_d$. This means that if we remove $x_d$'s assignment from both $A_i^{(y)}$ and $A_j^{(\neg y)}$ (creating $A_i^{(y)}\backslash x_d$ and $A_j^{(\neg y)}\backslash x_d$), then knowing $\left((A_i^{(y)}\backslash x_d) \wedge (A_j^{(\neg y)}\backslash x_d)\right)$ is true means knowing $x_d$'s value is sufficient to determine whether $y$ or $\neg y$ is true,

$$Known(x_d) \wedge (A_i^{(y)}\backslash x_d) \wedge (A_j^{(\neg y)}\backslash x_d) \implies Known(y).$$

We conduct further checks (see Appendix A.1.2) to ensure that the assignments themselves do not already imply a value for $y$,

$$(A_i^{(y)}\backslash x_d) \wedge (A_j^{(\neg y)}\backslash x_d) \implies \neg Known(y),$$

discarding any assignment from that doesn't satisfy the above property. We define $\tilde{\mathcal{A}}$ as the set of assignments satisfying the two properties above, and $\tilde{A}$ as an element of this set.

$$
\begin{aligned}
\tilde{\mathcal{A}} = \big\{ (A_i^{(y)}\backslash x_d) \wedge (A_j^{(\neg y)}\backslash x_d) : \\
\exists x_d \in X, \\
\left( Known(x_d) \wedge (A_i^{(y)}\backslash x_d) \wedge (A_j^{(\neg y)}\backslash x_d) \implies Known(y) \right) \\
\wedge \left( (A_i^{(y)}\backslash x_d) \wedge (A_j^{(\neg y)}\backslash x_d) \implies \neg Known(y) \right) \big\}
\end{aligned}
\tag{1}
$$

where the sufficient set of each $\tilde{A} \in \tilde{\mathcal{A}}$ is defined as

$$
\mathcal{C}(\tilde{A}) = \left\{ x : \left( \text{Known}(x_d) \wedge \tilde{A} \implies \text{Known}(y) \right) \wedge \left( \tilde{A} \implies \neg \text{Known}(y) \right) \right\}.
\tag{2}
$$

### A.1.1   CONSTRUCTING ALL ASSIGNMENTS THAT IMPLY $y$ IS TRUE

We begin by doing backwards search through the constraints $C$ to iteratively derive the set of assignments which imply $y$ is true. At each iteration, we keep track of a disjunction of conjunctions of variables that must be set in order for the goal variable to be true. We start from the most trivial assignment – just setting goal variable itself $y$ to true.

$$g_0 = y$$

We then find all rules which imply $y$ is true and add the relevant variable assignments to our set. For example, suppose we have rule $a \wedge b \to y$ and $c \wedge \neg d \to y$, then at the first iteration, we expand our disjunction of conjunctions to include

$$g_1 = (a \wedge b) \vee (c \wedge \neg d)$$

We then iterate through each conjunction, and apply the above procedure on each variable in the conjunction. For example, if we have $e \wedge f \implies b$ and $e \wedge g \wedge h \implies b$ and $e \wedge x \implies c$ in the constraints, then we expand as:

$$g_2 = ((a \wedge ((e \wedge f) \vee (e \wedge g \wedge h)))) \vee ((e \wedge x) \wedge \neg d)$$

Which we can then apply the distributive property to, obtaining

$$g_2 = (a \wedge e \wedge f) \vee (a \wedge e \wedge g \wedge h) \vee (e \wedge x \wedge \neg d)$$

More formally, we decompose each constraint $C_i$ into $\text{PREMISES}(C_i) \implies \text{CONCLUSION}(C_i)$[8], where $\text{PREMISES}(C_i)$ is a set of terms that participate in the conjunction, while $\text{CONCLUSION}(C_i)$ is a single term implied by $\text{PREMISES}(C_i)$. Backwards search is thus formalized as follows:

$$g_0 = y$$

$$g_1 = \bigvee_i^{[N]} \left( \bigwedge_j^{[M_i]} t_{i,j} \right), \ \forall t_{i,j} \in \text{PREMISES}(C_i), \forall C_i \text{ where } \text{CONCLUSION}(C_i) = y$$

$$g_2 = \bigvee_i^{[N]} \left( \bigwedge_j^{[M_i]} \left( \bigvee_k^{[N_j]} \left( \bigwedge_\ell^{[M_k]} t_{i,j,k,\ell} \right) \right) \right), \ \forall t_{i,j,k,\ell} \in \text{PREMISES}(C_{i,j,k}),$$

$$\forall C_{i,j,k} \text{ where } \text{CONCLUSION}(C_{i,j,k}) = t_{i,j}, \cdots$$

$$= \bigvee_i^{[N]} \bigvee_{\substack{k', \\ \forall (j,k') \in ((0,k'),\cdots,(M_i,k')) \\ \forall ((0,k'),\cdots,(M_i,k')) \in \\ \bigtimes\{\{(j,k)\forall k\in[N_j]\}\forall j\in[M_i]\}}} \bigwedge_j^{[M_i]} \bigwedge_\ell^{[M_k]} t_{i,(j,k'),\ell}$$

$$\forall t_{i,(j,k'),\ell} \in \text{PREMISES}(C_{i,(j,k')}),$$

$$\forall C_{i,(j,k')} \in \bigtimes_j \{C_{i,j,k} \forall C_{i,j,k} \text{ where } \text{CONCLUSION}(C_{i,j,k}) = t_{i,j}\}$$

taking all combinations of $k$ rules that can form each $j$ term

$$= \bigvee_{i_2} \bigwedge_{j_2} t_{i_2, j_2} \qquad \text{re-indexing}$$

$$\cdots$$

to infer all sets of variable assignments that implies $y$. Similarly, we repeat the process starting from $\neg y$.

This gives us the full set of variable assignments $A^{(y)}$ which imply $y$ is true. We also repeat this backwards-search procedure starting from $\neg y$ to get the full set of variable assignments $A^{(\neg y)}$ which imply $\neg y$ is true.

### A.1.2 CHECKING 1-SUFFICIENCY

After constructing potential 1-sufficient assignments $A^{y,-1} = \{(A_i^{(y)} \backslash x_d) \wedge (A_j^{(\neg y)} \backslash x_d) \forall x_d \in X\}$, we conduct several further checks to ensure they are 1-sufficient:

1. First, we check that

$$\forall A_i^{y,-1} \in A^{y,-1},$$

$$\left( \forall A_i^{(y)} \in A^{(y)}, A_i^{y,-1} \centernot\implies A^{(y)} \right) \wedge$$

$$\left( \forall A_i^{(\neg y)} \in A^{(\neg y)}, A_i^{y,-1} \centernot\implies A^{(\neg y)} \right)$$

---

[8]Note that any rule of form $a \wedge b \wedge c \implies d$ is equivalent to $a \wedge b \wedge \neg d \implies \neg c, a \wedge \neg b \wedge c \implies \neg d$, etc. We consider all possible cycles by writing $C_i$ in the form of a disjunction, $\neg a \vee \neg b \vee \neg c \vee d$ and seeking all disjunctions that contain a particular term (e.g. $d$), meaning they are implied by the conjunction of the negation of the rest of the terms (e.g. $a \wedge b \wedge c$).

This ensures that $A_i^{y,-1}$ by itself is not sufficient to determine the value of $y$. We throw away any $A_i^{y,-1}$ that does not satisfy these two constraints.

2. For any $A_i^{y,-1} \in A^{y,-1}$, if exists another $A_{i'}^{y,-1} \in A^{y,-1}$ such that $A_i^{y,-1} \implies A_{i'}^{y,-1}$, then all variables in the sufficiency set of $A_{i'}^{y,-1}$ is also in the sufficiency set of $A_i^{y,-1}$. However, resolving $A_{i'}^{y,-1}$ may require shallower backwards search than resolving $A_i^{y,-1}$ (see Appendix C.1). To ensure that the LM conducts search to the full depth required for resolving $A_i^{y,-1}$, we construct an "*invalid set*" consisting of the elements of the resolution sets of $A_{i'}^{y,-1}$. During test-time, we tell the LM that it cannot ask about the value of any $x'_d$ in the sufficiency set of $A_{i'}^{y,-1}$.

## A.2 PLANNING-Q

We introduce some notation to describe the dataset construction procedure for Planning-Q. The fully-specified version of the Blocks World task with $n$ blocks can by characterized as a MDP $\langle X, S, Q, s_0, y \rangle$ where

- $X$ is the full set of atoms $p$ that can be true of a state.
- $S$ is the set of *physically-possible* fully-specified states, which can be represented as a full set of assignments from all variables in $X$ to $\{\text{TRUE}, \text{FALSE}\}$.
- $Q : S \to S$ is a set of actions that operate on the current state and transitions it to a next state. Each action $q \in Q$ has a set of preconditions which must hold for the action to be applicable in the state, and a set of effects which hold after the action is applied. Preconditions and effects can be expressed as a conjunction of atoms $x$ or negated atoms $\neg x$ for any $x \in X$. In the blocks setting, there are 4 types of actions which can be enacted on each block:

```
pick-up(?x)
    :precondition (and (clear ?x) (ontable ?x) (handempty))
    :effect (and (not (ontable ?x)) (not (clear ?x))
                (not (handempty)) (holding ?x))
put-down(?x)
    :precondition (holding ?x)
    :effect (and (not (holding ?x)) (clear ?x)
                (handempty) (ontable ?x))
stack(?x, ?y)
    :precondition (and (holding ?x) (clear ?y))
    :effect (and (not (holding ?x)) (not (clear ?y))
                (clear ?x) (handempty) (on ?x ?y))
unstack(?x, ?y)
    :precondition (and (on ?x ?y) (clear ?x) (handempty))
    :effect (and (holding ?x) (clear ?y) (not (clear ?x))
                (not (handempty)) (not (on ?x ?y))))
```

- $s_0$ is the initial state, which is in $S$.
- $y$ is a conjunction of goal propositions which we wish to be true at the end of a plan. There are 6 possible $y$'s in this dataset, which were written manually by the authors:

```
{(on b a),
 (and (on b a) (on c b)),
 (and (on b a) (on d c)),
 (and (on b a) (ontable a)),
 (and (on b a) (ontable a) (on c b)),
 (and (on b a) (ontable a) (on d c) (ontable c))}
```

Given the above, models are expected to construct an **optimal action sequence** $\hat{\tau} = [q_0, q_1, \cdots, q_{k-1}]$ where $q_t \in Q$ for all $q_t$ in the trajectory.

**Definition: (Optimal) Action Sequence.** An action sequence $\tau$ enables the robot to go from initial state $s_0$ to a goal state $s_k$ where the goal conditions are satisfied ($s_k \implies y$). We use TRANSITION($s_0, \tau$) to denote the state of the robot after taking action sequence $\tau$ from state $s_0$. We also use $s_0, \cdots s_k$ to denote the sequence of intermediate states the robot goes through when taking the action sequence, where action $a_i$ results in state $s_{i+1}$. An optimal sequence $\hat{\tau}$ is the shortest path

that satisfy TRANSITION$(s_0, \tau)$ implies $y$.

$$\hat{\tau} = \arg\min_{\tau} |\tau| \quad \text{s.t. TRANSITION}(s_0, \tau) \implies y \tag{3}$$

**Definition: Planner.** We use $\Psi_y$ to denote an optimal planner that can map initial states $s_0$ to an optimal actions sequence $\hat{\tau}$ to the goal $y$. The planner is implemented through breadth-first-search, on a search graph where the nodes are the states and the edges are the actions, starting from state $s_0$ and terminating in a state where $y$ is true.

In order to make the planning problem underspecified, we construct a version of it where certain predicates in $s_0$ are withheld, such that the initial state is only partially observed.

**Definition: Partial state.** A partial state is one where a subset of predicates have been assigned values.

**Definition: Consistent set.** We say a full state $s$ is consistent with a partial state $\tilde{s}$ if $s \in S$ and $s \implies \tilde{s}$. The set of all states which are consistent with $\tilde{s}$ is called the consistent set for $\tilde{s}$, which we denote with $F(s)$.

We begin by inferring all partial states $\tilde{s_0}$ where there is only a single possible optimal action sequence from any $s_0 \in F(\tilde{s_0})$ to the goal. In other words, if we know $\tilde{s_0}$, we know the optimal trajectory to the goal. This gives us all the fully-specified CSPs.

$$\tilde{\mathcal{S}}_0 = \{\tilde{s_0} : \exists \hat{\tau}, s_0 \in F(\tilde{s_0}) \implies \Psi(s_0) = \hat{\tau}\} = \{\tilde{s} : \tilde{s} \implies \text{Known}(\tau)\}. \tag{4}$$

The construction process for this set is given in Appendix A.2.1.

In order to construct 1-sufficient initial states, we withhold a single atom from each $\tilde{s_0} \in \tilde{\mathcal{S}}_0$ and check that knowing the truth value of a single additional atom in $X$ is *necessary and sufficient* in order to disambiguate **a single optimal action sequence** to the goal. Details are given in Appendix A.2.2.

$$\tilde{\mathcal{S}}_0{}' = \left\{\tilde{s_0}' : \exists x \in X, \left(\text{Known}(x) \wedge \tilde{s_0}' \implies \text{Known}(\tau)\right) \wedge \left(\tilde{s_0}' \implies \neg\text{Known}(\tau)\right)\right\} \tag{5}$$

where the sufficient set of each $\tilde{s_0}'$ is defined as

$$\mathcal{C}(\tilde{s_0}') = \{x : \left(\text{Known}(x) \wedge \tilde{s_0}' \implies \text{Known}(\tau)\right) \wedge \left(\tilde{s_0}' \implies \neg\text{Known}(\tau)\right)\}. \tag{6}$$

### A.2.1 DERIVING ALL FULLY-SPECIFIED PARTIAL STATES WITH THE SAME OPTIMAL ACTION SEQUENCE TO $y$

Given goal conditions $y$, first, we perform backwards breadth-first-search from $y$ to derive the full set of optimal partial-state trajectories that end at $y$.

**Definition: Partial-state Trajectories.** A trajectory $\tilde{\tau}$ where the intermediary states are partial states, e.g. $[\tilde{s}_0, q_0, \tilde{s}_1, q_1, \cdots, \tilde{s}_k]$, where $\tilde{s}_{1\ldots K}$ are partial states. Partial-state trajectories are valid if for any consecutive state sequence $\tilde{s}_t, q_t, \tilde{s}_{t+1} \in \tilde{\tau}$, applying $q_t$ to any full state $s_t \in F(\tilde{s}_t)$ arrives at some $s_{t+1} \in F(\tilde{s}_{t+1})$. They are optimal if the trajectory is the shortest trajectory from $\tilde{s}_0$ to $\tilde{s}_k$.

Starting from the atoms in $y$, we iteratively search each action and deduce the partial states from which applying that action result in $y$.

$$g_0 = \{[y]\}$$
$$g_1 = \{[\tilde{s}, q, y], \; \forall q \in Q \, \forall \tilde{s} \text{ s.t. TRANSITION}(\tilde{s}, q) \implies y\}$$
     *find all partial states $\tilde{s}$ that transition into $y$ on some action $q$,*
     *prepend $\tilde{s}, q$ to the existing trajectories*
     $\cdots$
$$g_{i+1} = \{[\tilde{s}, q, \tilde{\tau}], \; \forall q \in Q \, \forall \tilde{s} \text{ s.t. } \exists \tilde{\tau} \in g_i \text{ where TRANSITION}(\tilde{s}, q) \implies \tilde{\tau}[0]\}$$
     *find all partial states $\tilde{s}$ that transitions from some action $q$ into an initial state $\tilde{\tau}[0]$*
     *of a trajectory $\tilde{\tau}$ found in the prior iteration $g_i$. Prepend $\tilde{s}, q$ to that trajectory*
     $\cdots$

We expand a search tree where each branch of the tree is partial trajectory. We terminate search for that branch if we arrive at a partial state $\tilde{s}$ that is implied by a partial state we have already encountered (meaning we have already considered that partial state). This means in the worst case, we exhaust the space of all possible partial states. When all branches have terminated, we have the set of all optimal partial-state trajectories to the goal $y$, by taking the union $g_0 \cup g_1 \cup \cdots$. The set of fully specified partial states $\tilde{\mathcal{S}}_0$ is the initial states of all these trajectories.

### A.2.2 CREATING 1-SUFFICIENT PARTIAL STATES

To create 1-sufficient partial states from $\tilde{\mathcal{S}}_0$, we remove one proposition $x_d$ from each $\tilde{s}_0 \in \tilde{\mathcal{S}}_0$ to create $\tilde{S}'_0 = \{\tilde{s}_0 \backslash x_d \forall x_d \in \tilde{s}_0 \forall \tilde{s}_0 \in \tilde{\mathcal{S}}_0\}$ where $\tilde{s}_0' = \tilde{s}_0 \backslash x_d$ for some $\tilde{s}_0$, and check that each of the following holds:

1. For all other $\tilde{s}_0'' \in \tilde{\mathcal{S}}_0$, we check whether $\exists x \in X, \tilde{s}_0'' = \tilde{s}_0' \wedge x$, meaning that a single additional true atom brings us to another 1-sufficient partial state. In this case, we assign the salient atom $x$ where $\tilde{s}_0'' = \tilde{s}_0' \wedge x$ to FALSE, to eliminate the possibility more than one question must be asked: for example, if $x$ is asked and turns out to be true, then we are brought to $\tilde{s}_0''$, which we know is 1-sufficient, meaning we must ask at least one other question.
2. For all physically-valid, fully-specified states $s_0' \in F(\tilde{s}_0')$ consistent with the 1-sufficient state $\tilde{s}_0'$, we check that there are at most two possible distinct optimal action sequences to the goal condition $y$:

$$\left|\{\Psi_y(s_0'), \ \forall s_0' \in F(\tilde{s}_0')\}\right| \leq 2$$

   (a) If there is 1 unique action sequence, then we expect the LM response to be "*No questions needed.*"
   (b) If there are 2 unique action sequences $\tau_1, \tau_2$, then we separated out $\tilde{S}_0$ into $\tilde{S}_0^{(1)}$ and $\tilde{S}_0^{(2)}$, where the optimal action sequence from all states in $\tilde{S}_0^{(1)}$ to the goal is $\tau_1$, while the optimal action sequence from all states in $\tilde{S}_0^{(1)}$ to the goal is $\tau_2$ ($\tilde{S}_0 = \tilde{S}_0^{(1)} \cup \tilde{S}_0^{(2)}$). We find the set of differentiating attributes between $\tilde{S}_0^{(1)}$ and $\tilde{S}_0^{(2)}$, that is to say, all attributes of states in $\tilde{S}_0^{(1)}$ that aren't present in any state of $\tilde{S}_0^{(2)}$, or vice versa (all attributes of states in $\tilde{S}_0^{(2)}$ that aren't present in any state of $\tilde{S}_0^{(1)}$). If any of these questions are asked, they would disambiguate whether the optimal action sequence is $\tau_1$ or $\tau_2$.

This ensures that for all $\tilde{s}_0'$, asking about the truth value of $x_d$ fully determines a unique optimal trajectory from the state to the goal conditions. If $\tilde{s}_0'$ passes all of the above checks, we add it to the set of 1-sufficient partial states $\tilde{\mathcal{S}}_0'$

### A.3 GSM-Q

Full instructions we provided to annotators can be found below:

You will be presented with a series of math problems. These math problems are written in words and may be ambiguous. Your task is to try to solve the problem. The problem may be ambiguous, which would make it unsolvable. However, if the problem is solvable, you will be asked to provide the answer, and may additionally be asked to translate the problem into a set of variables and equations given the information present in the problem. Two examples are provided below. Please read carefully and make sure you understand before proceeding.

Math problem 1:
*If there are 10 eggs in a basket. Alice buys more eggs and increases her egg quantity by 200%, but she had also sold half of her eggs by then. How many eggs are there total?*

You will be asked to try and solve the problem to check if it is ambiguous.
1. Try to solve this problem. What is the answer?: [text box]
   ☐ Not sure. Explain why: [text box E]
      What questions, when answered, could clarify this problem?: [text box Q]
In this case, the problem is ambiguous. You should check off "Not sure" and write why the problem is ambiguous in the explanation text box E. For example, in this case, you may write: it is unclear whether "increases by 200%" means 200% or 300% of her original amount. Furthermore, it is unclear which came first: did she sell half her eggs before increasing by 200%, or did she buy 200% more eggs first, then sell half her eggs.

Next, you should write some questions that could be asked to clarify this problem in text box Q. For example, you may write "does an increase by 200% mean 200% or 300% of the original amount?", "which happened first, Alice buying more eggs or Alice selling half her eggs?"

Here are some other examples of ambiguous questions that raters have found in this dataset. **Note**: there may be some subjectivity when deciding whether or not a particular problem is ambiguous. Please base it off your own interpretation.

| Problem | Explanation |
|---|---|
| Janet buys a brooch for her daughter. She pays $500 for the material to make it and then another $800 for the jeweler to construct it. After that, she pays 10% of that to get it insured. How much did she pay? | The antecedent of "that" in "10% of that" is unspecified. |
| Josh decides to try flipping a house. He buys a house for $80,000 and then puts in $50,000 in repairs. This increased the value of the house by 150%. How much did he make? | What should be considered the initial value of the house is unclear. It could be taken as the initial purchase price or the initial purchase price plus repairs. Furthermore, it is unclear whether "increase by 150%" means 150% or 250% or the price. |
| Jason has a phone plan of 1000 minutes per month. Every day he has a 15-minute call with his boss, and he's had 300 extra minutes of call this month to other people. How many minutes does Jason have left if this month has 30 days? | The day of the month is not specified. |
| In a 60-item quiz, 40% of the questions are easy, and the rest are equally divided as average and difficult questions. If Aries is sure to get 75% of the easy questions, and half of the average and difficult questions correctly, how many points is she sure to get? | The number of points per question is not specified. They could all be worth one point or they could be weighted differently. |
| Mara added 3 slices of cake to a plate that already had 2 slices on it. She was getting hungrier so she tripled the number of slices she currently has. She ate 2 slices and while she was distracted, her friend stole 5 slices off her plate. What number of cake slices remained on the plate? | The order of events is unclear. Because of the temporal mismatch between "was getting" and "currently has" in the same sentence, a reader cannot know whether "the number of slices she currently has" refers to the number of slices before or after adding the 3. |

**Note 2**: the problem may be ambiguous in more ways than one. Please explain all ways the problem is ambiguous.

Math problem 2:
*If there are 10 eggs and 2 in a basket, and there are twice as many eggs in a second basket, how many eggs are there total?*

1. Try to solve this problem. What is the answer?: [text box]
   ☐ Not sure. Explain why: [text box E]
      What questions, when answered, could clarify this problem?: [text box Q]

In this case, you should answer 30. This is because there are 10 eggs in the first basket + 20 in the second basket, so 30 total. You should not check off "Not sure."

In cases where you did not check off "Not sure", you may be additionally asked to translate the problem into a series of equations, together with the variables that appear in the equations and the goal of the problem.

2. Please translate the above math problem into a list of variables, a list of equations, and a goal variable. Ensure that your translation is equivalent to the variables above.

For example, in this case, the problem may be translated as follows:

Variables:
A = 10 [Number of eggs in the first basket]
B [Number of eggs in the second basket]
T [Total number of eggs]

Equations:
B = 2 * A [There are twice as many eggs in the second basket as the first.]
T = A + B [The total number of eggs is the sum of the eggs in the first and second baskets.]

Goal: T. How many eggs are there total?

If the value of a variable is directly mentioned in the math problem, you should write down the value. For example, A = 10. If the value of a variable is not directly mentioned in the math problem, you should not write down the value, even if it can be computed. Here are some examples of incorrect translation:

INCORRECT EXAMPLE 1

      Variables:
      A = 10 [Number of eggs in the first basket]
      T [Total number of eggs]

      Equations:
      T = A + B. The total number of eggs is the sum of the eggs in the first and second baskets.

      Goal: T. How many eggs are there total?

The above example is missing a variable (the number of eggs in the second basket) and an equation.

INCORRECT EXAMPLE 2

      Variables:
      A = 10 [Number of eggs in the first basket]
      B = 2 [Number of eggs in the second basket]
      T = Total number of eggs

      Equations:
      T = A + B. The total number of eggs is the sum of the eggs in the first and second baskets.

      Goal: T. How many eggs are there total?

The above example has a wrong variable value and missed an equation. "There are twice as many eggs in the second basket as the first" should be translated into B = 2 * A instead of B = 2.

INCORRECT EXAMPLE 3

      Variables:
      T [Total number of eggs]

      Equations:
      T = 10 + 20. The total number of eggs is the sum of the 10 eggs in the first basket and the 20 eggs in the second basket.

      Goal: T. How many eggs are there total?

The above example combines too many operations into a single equation, in a way that is not faithful to the original question. A good rule of thumb is to have one variable stand in for every number in the problem, and have all equations be of one of two forms: (1) assigns one variable to one constant, or (2) assigns one variable to a relation among other variables. Avoid equations that can be simplified.

A screenshot of the annotation interface for each math problem can be found in Figure 3.

## B  LM PROMPTS FROM EACH DATASET

### B.1  LOGIC-Q

Suppose you know the following rules about Alice:
[rules_nl]
You trying to discern whether a statement about Alice is true given some facts. You must decide whether you have enough information to determine whether the final statement is true. You may respond with one of the following-
If you do not have enough information yet, you may ask a question about an attribute of Alice, in the form of "Question: Is Alice [attribute]?". Ask the best question that, regardless of how it is answered, provides the most information about the final statement.
Once you have enough all information necessary to determine the truth value of the statement, you can terminate with "End questioning".
Generate one of "Question: Is Alice [attribute]?" or "End questioning" and nothing else.

**Math Problem**

Carrie is planning the caroling schedule for the Christmas event. The choir plans to sing "Deck the Halls" twice, "Jingle Bells" once, and "Silent Night" three times. If "Deck the Halls" is 150 seconds long, "Jingle Bells" is 240 seconds long, and "Silent Night" is 180 seconds long but they decided not to sing it due to time constraints, how long will they be caroling?

**Evaluation Criteria**

- Try to solve this problem. What is the answer?

540     **Proceed To Translation**

☐ Not sure

Please translate the above math problem into a list of variables, a list of equations, and a goal variable.

**List of variables in the above problem**

| Variable name (e.g. B) | Variable value (e.g. 10, or leave it empty if the value is not directly mentioned) | Variable definition (e.g. "number of eggs in the first basket") | + |
|---|---|---|---|
| |      +   - | | x |

**List of equations in the above problem**

All equations must be a combination of the following and **no other symbols or characters**

1. The variables defined above
2. Integers and decimals
3. Operations, as standardized below:
   * for times (multiply)
   + for plus
   - for minus
   / for divides
   = for equals
   () for parentheses
   You may use more operations than the ones defined above if needed, but if the operation you need is defined above, please use the appropriate symbol.
4. Do NOT include any units ($, hours, days, etc.)
5. Remember to include * any time you are performing multiplications, e.g. use 2 * A and not 2A.
6. Convert all percentages into decimals, e.g. 150% becomes 1.5

| Equation (e.g. T = A + B) | Equation verbal description (e.g. "The total number of eggs is the sum of the number of eggs in the first basket and the number of eggs in the second basket.") | + |
|---|---|---|
| | | x |

**Goal variable:**

| Goal variable (e.g. T) | Goal description (e.g. "how many eggs are there total?") |
|---|---|
| | |

**Submit**

Figure 3: Screenshot of the annotation interface used for obtaining CSPs for each math problem in the GSM setting.

## B.2 PLANNING-Q

You will be given a planning problem in the domain defined by the following PDDL:
;;;;;;;;;;;;;;;;;;;;;;;;;;;;;;;;;;;; ;;; 4 Op-blocks world ;;;;;;;;;;;;;;;;;;;;;;;;;;;;;;;;;;;;
(define (domain BLOCKS)
(:requirements :strips :typing) (:types block) (:predicates (on ?x - block ?y - block) (ontable ?x - block)
(clear ?x - block) (handempty) (holding ?x - block) )
(:action pick-up :parameters (?x - block) :precondition (and (clear ?x) (ontable ?x) (handempty)) :effect
(and (not (ontable ?x)) (not (clear ?x)) (not (handempty)) (holding ?x)))
(:action put-down :parameters (?x - block) :precondition (holding ?x) :effect (and (not (holding ?x))
(clear ?x) (handempty) (ontable ?x))) (:action stack :parameters (?x - block ?y - block) :precondition
(and (holding ?x) (clear ?y)) :effect (and (not (holding ?x)) (not (clear ?y)) (clear ?x) (handempty)
(on ?x ?y))) (:action unstack :parameters (?x - block ?y - block) :precondition (and (on ?x ?y) (clear
?x) (handempty)) :effect (and (holding ?x) (clear ?y) (not (clear ?x)) (not (handempty)) (not (on ?x
?y)))))
The current objects are present in the problem:
['a', 'b', 'c', 'd', 'e']
You will be given a set of conditions true in your initial state and a set of goal conditions, and will
need to construct a plan from your current state to the goal state. Some details of your initial state
may be missing. You must decide whether you have enough information to disambiguate a plan to the
final state. If not, you must decide what information is necessary to construct a fully unambiguous plan
from your initial state to the goal state. You will be presented with a set of multiple-choice options for
questions you may ask, and you must answer with one of the options. Please reason step-by-step, then
generate "Answer:" followed by the number of the option and nothing else.
Known facts about current state:
[current_state]
Goal state:
[goal_state]
Possible questions:
0. Is (clear a) true?
1. Is (clear b) true?
2. Is (clear c) true?
...

## B.3 GSM-Q

□

# C  SEARCH SOLUTIONS FOR LOGIC-Q AND PLANNING-Q

## C.1  LOGIC-Q

**Brute-force Solution.**  A problem in Logic-Q can be solved through brute force search as follows.
We have a subroutine `infer` that allows us to infer the values of all variables consistent with the
current variable assignments. For example, if $A = \{a, b\}$ and we know that $a \wedge b \to c$, then
`infer(A)` gives us $\{a, b, c\}$. At a high-level, `infer` is analogous to breadth-first-search, and thus
has runtime complexity $O(|X| + |C|)$.

To solve a Logic-Q problem, we first run `infer` to get values of all variables consistent with the
current assignment $A$, creating $A'$. Next, for all unassigned variables $x_u \in X \backslash A'$, we iterate through
them one at a time and check whether:

1. Including $x_u$ in $A'$ allows us to infer the target variable is either *true* or *false*.
2. Include $\neg x_u$ in $A'$ allows us to infer the target variable is the *opposite assignment* as it was
   in case (1).

The first time 1 and 2 both hold, we have that $x_u$ is true.

The runtime of brute-force can thus be computed by the expected number of variables that we
need to iterate through to get to a variable in the sufficient set (denote this $\mathbb{E}_{\text{BF}}$), multiplied by
the complexity of running the `infer` algorithm twice. By Markov's inequality, with probability

$\geq 1 - \delta$, the complexity is bounded by

$$O((|X| + |C|)\mathbb{E}_{\text{BF}}/\delta)$$

**Backwards Search Solution.**  A problem in Logic-Q can be solved through backwards search. The procedure is the same as the backwards search used to construct the dataset, described in Appendix A.1.1. At each iteration, we keep track of a disjunction of conjunction of variables required to prove $y$.

The complexity of backwards search is given by the search depth $d$ multiplied by the number of expansions per depth, which is bounded by $O(3^{|X|}|C|)$ (expanding up to $|C|$ rules for conjunction, for which there are at most $3^{|X|}$ conjunctions – each variable can be either true/false/missing from that conjunction) Thus, the overall complexity is $O(3^{|X|}|C|d)$.

Though comprehensive backwards search is EXP-time, we can terminate early as soon as we find a disjunction consisting of the negation of all initial conditions, the goal variable, and an additional term, which would mean that asking about the value of the initial term is sufficient to infer the value of the goal variable.

If we know this disjunction is at most at depth $d$, then the runtime is bounded by $|X|^{|C|d}$, where $|X|^{|C|}$ is the branching factor at each node. The branching factor comes from the cross product of at most $|X|$ terms across at most $|C|$ conjunctive constraints. Thus, the total runtime is given by

$$O(|X|^{|C|d})$$

## C.2   PLANNING-Q

**Brute-force Solution.**   Given an underspecified initial state $\tilde{s}_0$, we can generate all physically-plausible initial states $s_0 \in F(\tilde{s}_0)$. We can then search *all* initial states in this set, and partition the consistent initial states based on their optimal plan. This is analogous to the procedure used to construct 1-sufficient partial states for this dataset, described in Appendix A.2.2.

1. Using breadth-first search, we find the shortest path $\hat{\tau}$ from the initial state $s_0$ to the goal condition.
2. We add $s_0$ to the partition corresponding to $\hat{\tau}$.

The dataset was constructed so there is at most 2 partitions. At the end of this process, we examine each partition and extract the attributes, common to the initial states of a partition, that do not appear in the initial states of the other partition. Knowing the value of these variable allow us to distinguish which partition we are in, and accordingly which optimal plan takes us to the goal.

Breadth-first search takes at most $O(|S| + |S||\mathcal{A}|)$ time, and we perform Breadth-first search at most $|S|$ times, meaning the overall runtime of this solution is $O((|S| + |S||\mathcal{A}|)|S|)$. The number of states is bounded by $|S| = 2^{|X|}$, while the number of actions is bounded by $|\mathcal{A}| = 2b + 2b(b+1) = O(b^2)$ where $b$ is the number of blocks in the domain, so the overall complexity is:

$$O(2^{2|X|}b^2)$$

**Backward Search Solution.**   We replace the breadth-first-search from each consistent state with a single backwards search. Starting from the goal condition, we iterate backwards to find sets of initial states that utilize the same path to the goal. This is the same backwards search that was used to construct all fully-specified partial states, described in Appendix A.2.1. We expand backwards until we arrive at a set of partial initial states that are consistent with the given partially-observed set $\tilde{s}_0$, and we find the attribute that distinguishes each partial initial state from each other.

Because the number of partial states is bounded by $3^{|X|}$ (each proposition can take on 3 values: true/false/unknown), the backwards breadth-first search takes $O(3^{|X|} + 3^{|X|}|\mathcal{A}|)$. However, if we know that the longest path from any initial state consistent with $\tilde{s}_0$ to the goal is of length $d$, meaning we only need to search up to depth $d$, then the runtime is bounded by

$$O(|\mathcal{A}|^d) = O(b^{2d}) \text{ or } O(3^{|X|d})$$

(Each iteration, expand at most $3^{|X|}$ states or $|\mathcal{A}|$ actions.)

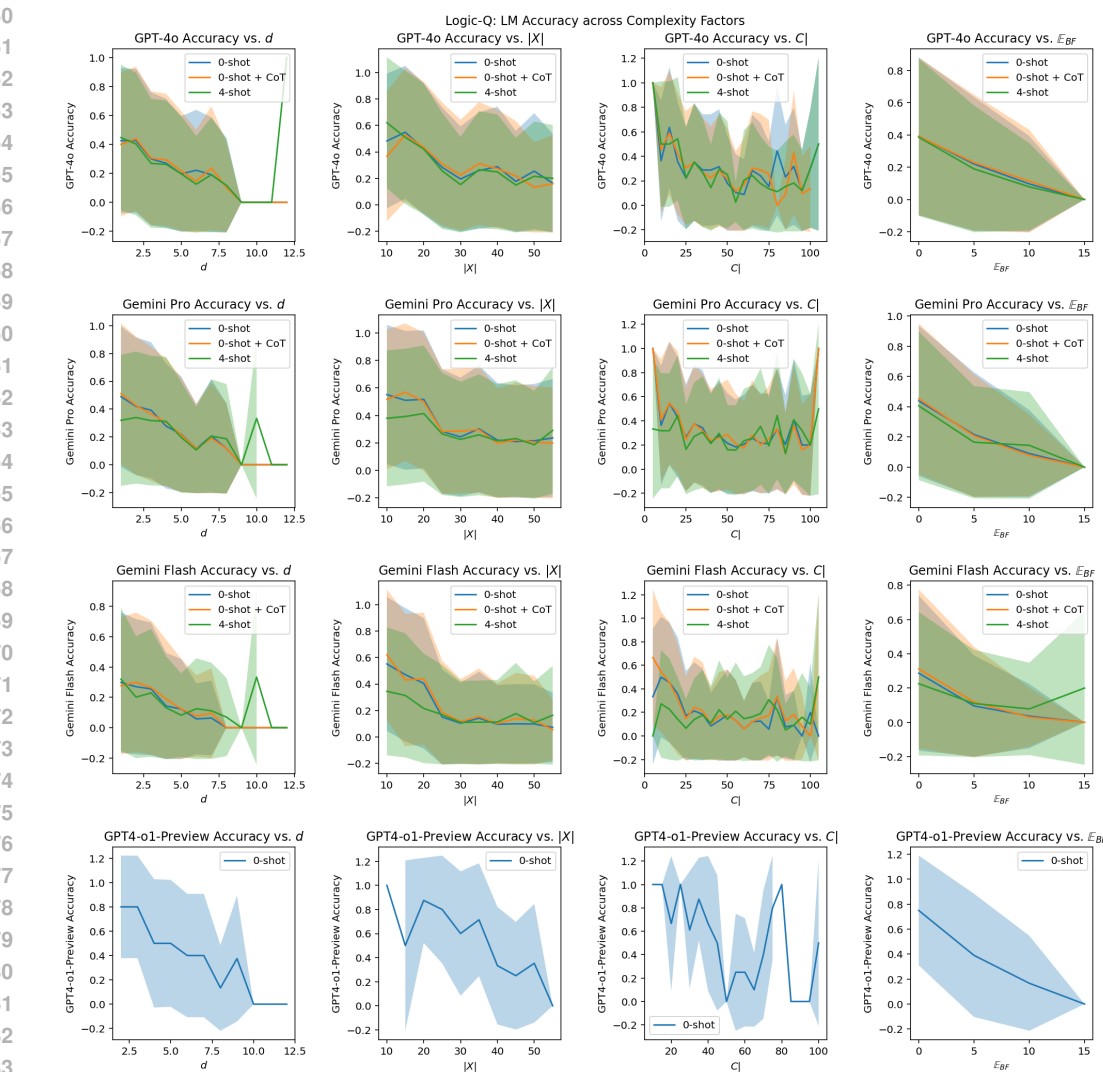

Figure 4: LM accuracies across varying backwards search depths $d$, number variables $|X|$, number constraints $|C|$, and expected number brute-force guesses $\mathbb{E}_{\text{BF}}$ for Logic-Q.

## D  PLOTS OF CORRELATIONS BETWEEN SEARCH COMPLEXITY AND LM ACCURACY

Plots of accuracy across each factor identified in Section 6 can be found in Figures 4 to 6.

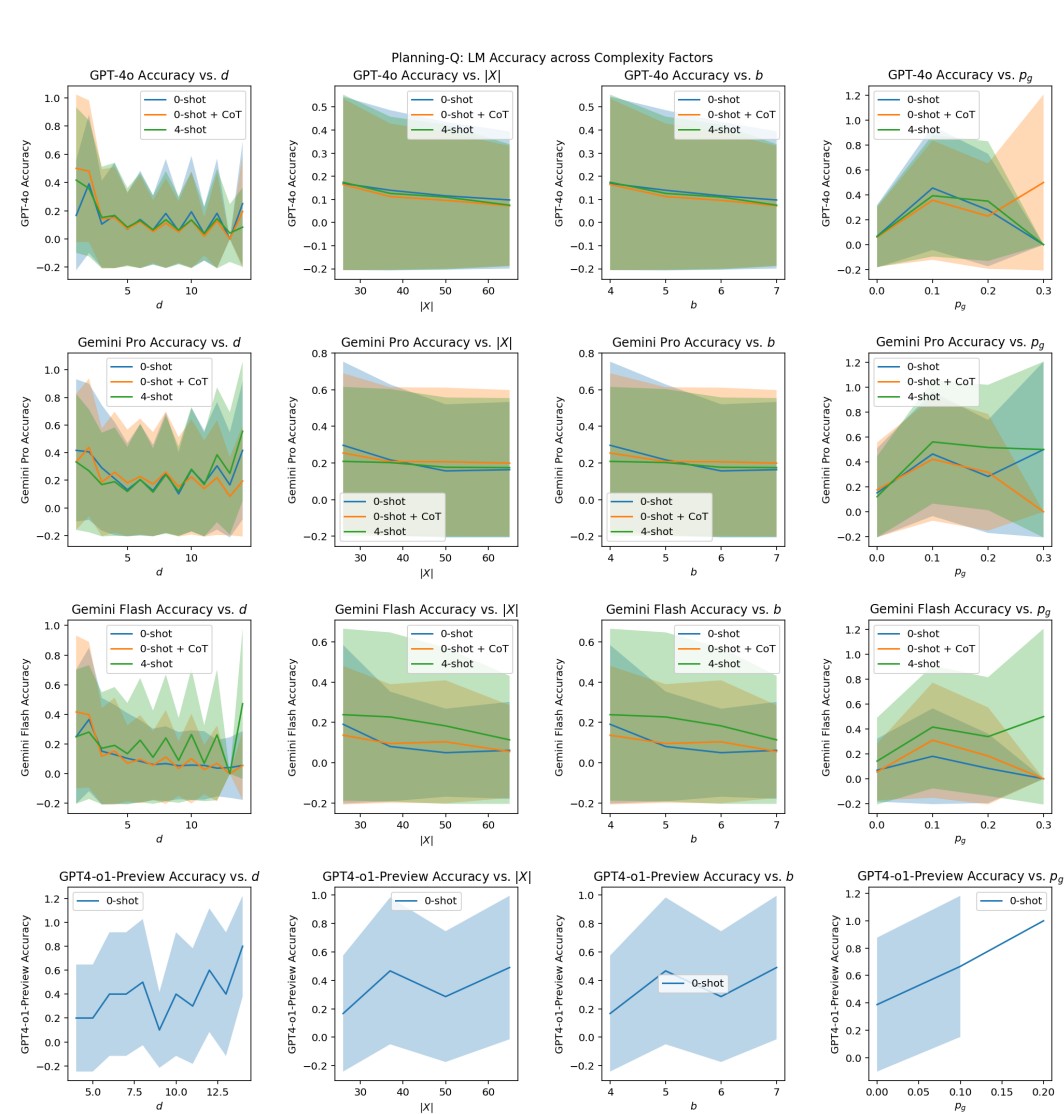

Figure 5: LM accuracies across varying backwards search depths $d$, number variables $|X|$, number blocks $b$, and random-guess correctness probabilities $p_g$ for Planning-Q.

1458
1459
1460
1461
1462
1463
1464
1465
1466
1467
1468
1469
1470
1471
1472
1473
1474
1475
1476
1477
1478
1479
1480
1481
1482
1483
1484
1485
1486
1487
1488
1489
1490
1491
1492
1493
1494
1495
1496
1497
1498
1499
1500
1501
1502
1503
1504
1505
1506
1507
1508
1509
1510
1511

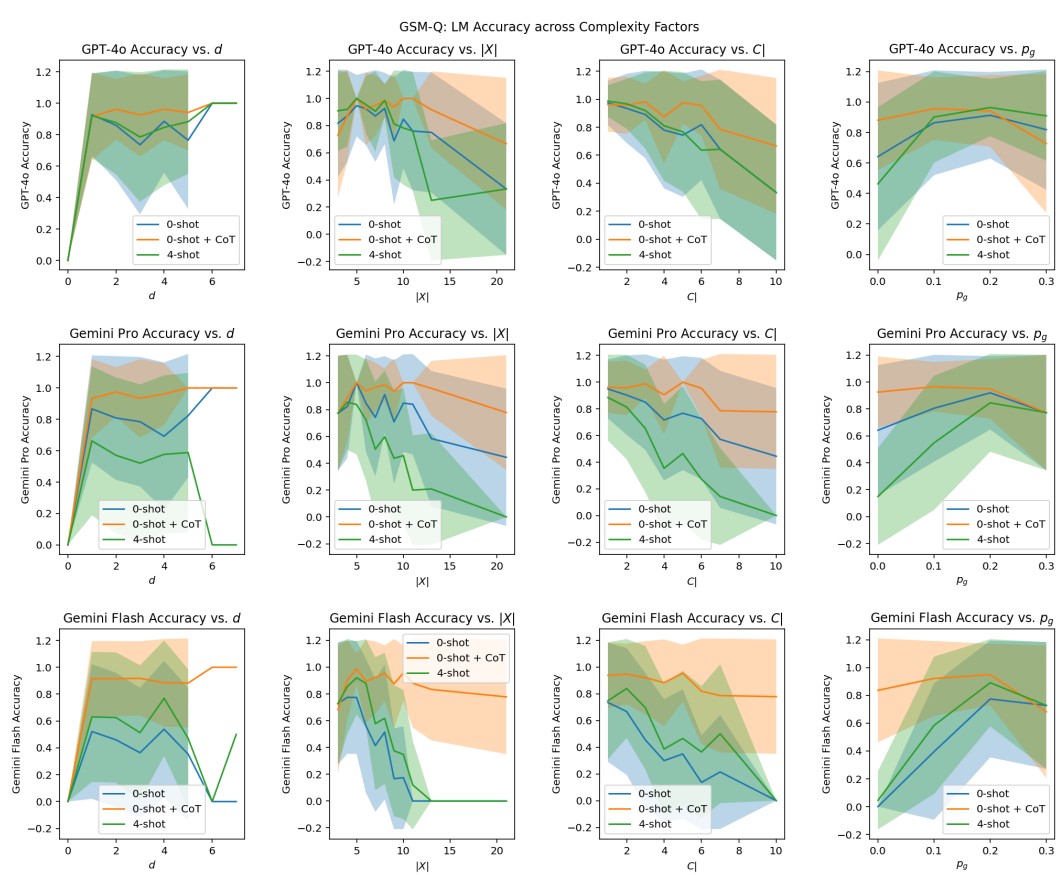

Figure 6: LM accuracies across search depths $d$, number variables $|X|$, number constraints $|C|$, and random-guess correctness probabilities $p_g$ for GSM-Q.

