# OpenReview forum: "Evaluating Information Gathering Abilities of Large Language Models with QuestBench"
_ICLR.cc/2025/Conference — Submitted to ICLR 2025_

### Official Review · Reviewer_ai7d · 2024-10-20

**Soundness:** 3
**Presentation:** 2
**Contribution:** 2
**Rating:** 3
**Confidence:** 3

**Summary:**

This paper focuses on the ability of large language models to actively request information from users when faced with semantically clear but underspecified questions. To evaluate this ability, the authors created QuestBench, a benchmark of underspecified tasks that can be solved by asking at most one question. This dataset includes three tasks:

- Logic-Q: Logical reasoning tasks where one proposition is missing
- Planning-Q: PDDL planning problems where the initial state is underspecified
- GSM-Q: Grade school math problems where one variable assignment is missing

The GSM-Q task was manually annotated. The authors evaluated existing models such as GPT-4o, Gemini, and o1. They found that even o1, which has significantly better reasoning abilities, struggles to perform well on these tasks.
This research highlights the challenges large language models face when dealing with underspecified questions and their ability to ask for clarification.

**Strengths:**

1. The information-gathering ability is important for large language models. The authors provide a dataset to evaluate this capability.
2. The definitions of key concepts and the methods for constructing the dataset are described very clearly.
3. The authors evaluated several advanced models (GPT-4o, Gemini, o1) and conducted some correlation analyses between search complexity and LLM accuracy.

**Weaknesses:**

1. While some models were evaluated, there was a lack of valuable findings and insights. Specifically,
- What are the potential reasons why the existing model lacks the ability of "information gathering"? Is it data, algorithm, or other factors?
- In what direction should we work further to improve the model's ability in this aspect?
- For the failure cases of the models, we can add some statistical analysis to summarize the types and causes of failures.

Thus, I suggest adding experiments about the following points that could be helpful:
- Select a subset of failure cases, summarize the reasons for model failure, and analyze the reasons that led to the failure.
- Discuss ways to improve the model's information-gathering capability. If possible, it would be better to conduct experiments to verify the feasibility of the methods.

2. There are important details missing in the evaluation process. Specifically, assessing the model's accuracy is a non-trivial task. This is because the correct behavior is to request the missing information in the underspecific question. However, the authors don't seem to describe this point in the paper. Overall, the author needs to provide details regarding how to judge whether the question asked by the model is correct.

**Questions:**

Could you please tell me how the accuracy was evaluated in this paper? Was it evaluated manually by humans or using an LLM? What were the specific evaluation criteria?

Could you please share some insights on how to improve the information-gathering capability of the model based on the evaluation results?

---

> ### Author Response · Authors · 2024-11-26
>
> > valuable findings and insights
>
> We analyzed the correlation between search complexity and accuracy. We believe that this is a valuable finding and insight.
>
> > why the existing model lacks the ability of "information gathering"? Is it data, algorithm, or other factors?
>
> We believe that there are multiple dimensions of reasons including
>
> - the lack of training or fine-tuning data for information gathering and question-asking tasks, especially for tasks that require asking a single best correct question;
> - the lack of planning and complex reasoning capabilities, as demonstrated in [1].
>
>
> > In what direction should we work further to improve the model's ability in this aspect?
>
> Please see the general reply “future work on method”.
>
>
> > For the failure cases of the models, we can add some statistical analysis to summarize the types and causes of failures.
>
> In our statistical analyses (Section 6), we found that the increase of search complexity leads to a growth in the number of failure cases. The types and causes are shown quantitatively in the factors described in Section 6.
>
> Could you please clarify what other types and causes we can include?
>
> > Select a subset of failure cases, summarize the reasons for model failure, and analyze the reasons that led to the failure.
>
> We have already done this kind of analysis in Section 6 and found the model failures are related to increase of search complexity. Please let us know if you have recommendations for other analyses.
>
> > Discuss ways to improve the model's information-gathering capability. If possible, it would be better to conduct experiments to verify the feasibility of the methods.
>
> Please see the general reply “future work on method”. Unfortunately given the limited space of the paper, we found it difficult to include methods.
>
> > details missing in the evaluation process. Specifically, assessing the model's accuracy is a non-trivial task
>
> Please see the general reply “how to compute accuracy”.
>
> > Could you please tell me how the accuracy was evaluated in this paper? Was it evaluated manually by humans or using an LLM? What were the specific evaluation criteria?
>
> Please see the general reply “how to compute accuracy”. We specifically avoid the need for humans or LLMs to evaluate since they can be unreliable. The tasks are multi-choice problems which makes it very easy to evaluate.
>
> > Could you please share some insights on how to improve the information-gathering capability of the model based on the evaluation results?
>
> Please see the general reply “future work on method”.
>
> [1] ​​Valmeekam, Karthik, et al. "On the planning abilities of large language models-a critical investigation." Advances in Neural Information Processing Systems 36 (2023): 75993-76005.

---

### Official Review · Reviewer_Rppr · 2024-11-01

**Soundness:** 2
**Presentation:** 2
**Contribution:** 2
**Rating:** 5
**Confidence:** 3

**Summary:**

The paper investigates the capabilities of LLMs to ask clarifying questions when dealing with underspecified tasks. These tasks often lack sufficient information to generate an accurate response without additional clarification. To evaluate this, the authors introduce QuestBench, a benchmark of three tasks (Logic-Q, Planning-Q, and GSM-Q) that require one clarifying question to resolve underspecified queries.

The study tested several SOTA models on QuestBench and found performance to be suboptimal. The findings reveal a gap in LLMs' ability to gather necessary information, particularly for complex logic and planning tasks.

The authors contribute by presenting a constraint satisfaction framework focused on evaluating underspecification and perform analyses on model performance correlations with reasoning mechanisms. Their results suggest LLMs struggle with larger solution spaces and deeper search requirements, showing potential limitations in the models' reasoning capabilities and highlighting areas for future improvements in question-asking and information-gathering skills in LLMs.

**Strengths:**

1. The paper presents a benchmark specifically aimed at evaluating the information-gathering abilities of LLMs when faced with underspecified tasks. The benchmark is well-designed and covers three types of tasks, including logic reasoning, planning, and math problems.

2. The paper provides various evaluations and insights into the types of reasoning mechanisms LLMs may currently lack, which could be useful in future improvements of the models.

**Weaknesses:**

1. The tasks in QuestBench are constructed to be solvable with only a single missing piece of information, which simplifies the challenges of real-world queries. This limited complexity limits the benchmark's applicability to real-world scenarios.

2. A potential weakness of the paper is the lack of natural language tasks in QuestBench. The current benchmark primarily includes structured tasks, such as logic reasoning, planning, and math problems, which lack the variability and richness of natural language interactions. This limits QuestBench’s ability to evaluate LLMs in more practical, real-world contexts where queries are often open-ended or conversational. Including natural language tasks would provide a more comprehensive assessment of LLMs’ information-gathering abilities, as these tasks better reflect the types of ambiguous and underspecified instructions encountered in everyday language use.

**Questions:**

1. In L415, why use four-shot settings.

---

> ### Author Response · Authors · 2024-11-26
>
> > The tasks in QuestBench are constructed to be solvable with only a single missing piece of information, which simplifies the challenges of real-world queries. This limited complexity limits the benchmark's applicability to real-world scenarios.
>
> Please see the general reply “Scope limited to 1-sufficient CSPs” and “applicability to real-world scenarios”.
>
> > lack of natural language tasks in QuestBench
>
> We asked human annotators to convert a subset of GSM-Q back into word problems (that are missing a premise), and evaluate how well GPT-4o is at finding the missing premise to ask about:
>
> * Original GSM-Q subset: 96.6% accurate
> * Verbalized GSM8K: 89.5% accurate
>
> Our instructions to human annotators for converting GSM-Q into word problems can be found below:
> > You will be presented with a series of math problems. These math problems are written in words and translated to equations. Your task is to first validate whether the translation is correct given the information present in the problem. If so, you will then be prompted to answer questions for each equation.
> > 1. Is the above list of variables, equations, and the goal equivalent to the original math problem written in words?
> > 2. Please solve for the “Goal” in the above list of variables and equations. Is your answer the same as [orig_answer]?:
> > 3. Try to rewrite the problem to remove all parts of the problem that states any of the above equation(s). Please make sure the problem is still coherent English (e.g. do not simply delete the section you copied above without fixing any grammatical errors). Please also make sure to remove the entire premise, not just replacing numbers with “few” or “some”. If there is no way to remove the equation (e.g. because it wasn’t mentioned in the original problem), please leave the text box empty and check off “cannot remove”.
> > 4. Given the above rewritten problem, is the answer to the question: [] the same as [orig answer], [] unclear, [] different from [orig_answer]
>
> Furthermore, as noted in Section 3, our work is much more focused on *underspecification* rather than *semantic ambiguity*. While we agree that every-day, natural-language tasks are rich with ambiguous instructions due to semantic ambiguity, we are focused specifically on cases when a piece of information is missing. Thus, we use CSPs as a jumping off point in order to disentangle these two types of ambiguity.
>
>
> > evaluate LLMs in more practical, real-world contexts where queries are often open-ended or conversational… ambiguous and underspecified instructions encountered in everyday language use
>
> Our goal is exactly NOT to evaluate queries that are open-ended since the ground truth is unclear and often subjective, which makes the evaluation unreliable. We specifically design the tasks to be 1-sufficient CSPs, so that there exists one correct question to be asked for each task. These are explained in the introduction (Section 1) and comparisons to prior work (Section 2), and our effort to disentangle ambiguity and underspecification (Section 3). To further clarify that we are not evaluating the generic information gathering skills of LLMs, we are changing the paper title to “QuestBench: Can LLMs ask the right question to acquire information in reasoning tasks?”.
>
> > In L415, why use four-shot settings.
>
> We wanted to benchmark standard setups of LLMs, and few-shot is one of the typical setup. We chose to use 4-shot because all tasks with 4-shot examples fit the context length of the models we evaluated. Other setups can be used but we believe the ones we evaluated are representative of the performance.

---

### Official Review · Reviewer_shNW · 2024-11-04

**Soundness:** 2
**Presentation:** 2
**Contribution:** 2
**Rating:** 5
**Confidence:** 3

**Summary:**

The paper introduces QUESTBENCH, a new benchmark designed to evaluate the ability of large language models (LLMs) to handle underspecified tasks by asking clarifying questions. QUESTBENCH frames these tasks as constraint satisfaction problems with missing information, focusing on scenarios where uncertainty arises due to missing variables rather than semantic ambiguity.
The benchmark consists of three categories:

1.	Logic-Q: Tasks involving logical reasoning where a missing proposition's value is needed.
2.	Planning-Q: Planning problems with undefined initial states requiring additional observations to reach a goal.
3.	GSM-Q: Grade school math problems lacking critical information for a solution.

The paper evaluates models like Gemini Pro 1.5, GPT-4o, and GPT-4-o1, finding significant room for improvement in their information-gathering abilities, with accuracy ranging from 20% to 44%.
Key contributions include:

1.	A framework for evaluating under-specification in LLMs.
2.	The creation of the QUESTBENCH benchmark for assessing LLMs' information-gathering skills.
3.	Analysis of LLM performance on QUESTBENCH, highlighting areas needing enhancement.

Overall, QUESTBENCH provides a structured approach to study how LLMs manage missing information and clarify underspecified instructions.

**Strengths:**

1. **Problem Formulation**: The paper introduces a novel benchmark, QUESTBENCH, designed to specifically assess the ability of LLMs to ask clarifying questions for underspecified tasks. This focus on missing information in constraint satisfaction problems distinguishes it from previous benchmarks. The use of constraint satisfaction as a method to frame underspecified tasks is an innovative approach, providing a structured way to evaluate models.
2. **Advancing LLM Capabilities**: By highlighting the current limitations of LLMs in handling underspecified tasks, the paper opens avenues for future research and development in enhancing model interactivity and problem-solving under uncertainty.
3. **Well-Defined Categories**: The definition of transforming Logic-Q, Planning-Q, and GSM-Q categories into a CSP problem is clear and logical, making the benchmark easy to understand.

**Weaknesses:**

1. **Limited Scope of Evaluation**:
* Details: While the paper evaluates several state-of-the-art models, it may benefit from testing a broader range of LLM families, including smaller or emerging models, to provide a more comprehensive understanding.
* Suggestions: Expand the evaluation to include open-source models like LLaMA, Qwen, Mistral, or close-sourced model like Claude 3.5
2. **The evaluation metrics are not clearly present**: In Table 2, the author mentions Language model accuracies at predicting the right question. But how do you define if a generated question is accurate?
3. **Lack of insights to handle underspecified problems**:
* First, the authors have shown numerous works that aim to actively seek clarification through questions, as noted in Lines 47-48. However, the author does not evaluate these methods, merely presenting the scores of some basic prompting strategies. Therefore, it is hard to say whether the low performance on QUESTBENCH is caused by inappropriate prompting.
* Second, this paper hardly shows the way to overcome the underspecified task. Though the major goal is this paper is for evaluation purpose, offering some insights into overcoming such a challenge could enhance its contribution

**Questions:**

1. In Table 2, how do you define if a generated question is accurate?
2. Can you show the results of the methods mentioned in Lines 47-48?

---

> ### Author Response · Authors · 2024-11-26
>
> > Limited Scope of Evaluation: Expand to other open-source LLMs
>
> We appreciate this suggestion. However, please note results on SOTA LLMs represent the upper bound on the information gathering ability of generic LLMs. Finding this upper bound helps us to answer our question on whether LLMs can actually ask the right question for information gathering, which is the goal of this work. Since this will be an open-sourced benchmark, people will be able to run any model on our benchmark if they are interested.
> That said, we will obtain more results in the new version of the paper.
>
> > The evaluation metrics are not clearly present
>
> Please see the general reply.
>
> > Lack of insights to handle underspecified problems: 1. …numerous works that aim to actively seek clarification through questions… does not evaluate these methods
>
> Please note that most of those works are designed for subjective or knowledge-based tasks such as persona-tasks “What is a good pasta recipe” [1], human preference eliciting tasks [2], knowledge-based ambiguity tasks “Who won the US open?” [3, 4] or knowledge-based medical diagnosis problems [5]. Their methods either do not apply to our tasks or require significant modifications (such as simulating users or designing rewards) to be applied to the underspecified reasoning tasks in our benchmark. We are not aware of existing methods that solve 1-sufficient CSPs defined in our work.
>
> > Lack of insights to handle underspecified problems: 2. …shows the way to overcome the underspecified task. Though the major goal is this paper is for evaluation purpose, offering some insights into overcoming such a challenge could enhance its contribution.
>
> Please see the general reply “future work on method”.
>
> > In Table 2, how do you define if a generated question is accurate?
>
> Please see the general reply “How to compute accuracy”.
>
> > Can you show the results of the methods mentioned in Lines 47-48?
>
> Please see the reply to “Lack of insights to handle underspecified problems: 1. …numerous works” above.
>
> [1] Chinmaya Andukuri, Jan-Philipp Franken, Tobias Gerstenberg, and Noah D Goodman. STaR-GATE: Teaching language models to ask clarifying questions. In Conference on Language Modeling, 2024.
>
> [2] Belinda Z. Li, Alex Tamkin, Noah Goodman, and Jacob Andreas. Eliciting human preferences with language models, 2023. URL https://arxiv.org/abs/2310.11589.
>
> [3] Michael JQ Zhang and Eunsol Choi. Clarify when necessary: Resolving ambiguity through interaction with LMs. arXiv:2311.09469 [cs.CL], 2023.
>
> [4] Jing-Cheng Pang, Heng-Bo Fan, Pengyuan Wang, Jia-Hao Xiao, Nan Tang, Si-Hang Yang, Chengxing Jia, Sheng-Jun Huang, and Yang Yu. Empowering language models with active inquiry for deeper understanding. arXiv preprint arXiv:2402.03719, 2024.
>
> [5] Zhiyuan Hu, Chumin Liu, Xidong Feng, Yilun Zhao, See-Kiong Ng, Anh Tuan Luu, Junxian He, Pang Wei Koh, and Bryan Hooi. Uncertainty of thoughts: Uncertainty-aware planning enhances information seeking in large language models. arXiv:2402.03271 [cs.CL], 2024.

---

### Official Review · Reviewer_Hjhv · 2024-11-04

**Soundness:** 2
**Presentation:** 2
**Contribution:** 2
**Rating:** 3
**Confidence:** 4

**Summary:**

Most practical problems require humans to operate in uncertain settings where uncertainty might arise from either ambiguity or underspecification of the problem. It then becomes imperative to obtain relevant information by asking clarification questions. While this is a common knowledge in human conversations, with the advent of LLMs, it is essential to evaluate their capacity to reason about uncertainty and actively acquire the necessary information for completing tasks. Existing benchmarks are limited in their scope and do not cover complex logic, planing, and math reasoning. The subjective nature of the problem where the information to acquire might vary based on individuals and population pose further challenges.
Towards that the authors present a collection of question asking benchmarks, that they call QuestBench, that cover logic, planning and grade school math. They specifically focus on problems that can be formulated as constrain specification problems (CSP) and, within that, limit the scope to problems that are underspecified. Their benchmark - QuestBench - leverages existing datasets - SimpleLogic, PyperPlan, and GSM-Plus and converts them into 1-sufficient CSP where the size of the smallest sufficient set (of variables required to solve the problem) is 1. They then evaluate some of the proprietary models on these benchmarks and find that while these models perform well in identifying missing information in GSM problems, they struggle with logic and planning problems. They also correlate this performance with different measures of search complexity and hypothesize that the LLMs might possess search skills similar to breadth-first search or brute-force approaches, which become less effective as the search space expands.

**Strengths:**

The ability of LLMs to ask clarification problems is important to advance their state-of-the-art and application to solving complex practical problems that are riddled with uncertainty. The authors attempt to address an important problem and take a step forward by presenting a quantitative benchmark to evaluate LLMs.
Through evaluation of top models, they highlight their inability in identifying missing information, especially in complex planning and logic reasoning tasks while doing relatively better on math. I also appreciate that the authors go a step further to correlate this performance with different measures of complexity. Their hypothesis around the limited search capability of current LLMs, derived from this correlation, seems intuitive and leads to a call for action for the LLM research community.

**Weaknesses:**

While the motivation is strong, by limiting the scope to 1-sufficient CSPs, I feel the scope is significantly limited. At least it is unclear how much of the practical problem space does this cover.
I also found some gaps in writing that make it difficult to follow. For instance, the challenges around identifying necessary information to acquire and the lack of ground truth would have been better understood with some examples. The notion of constraint satisfaction problems is loosely defined. It is unclear what class of practical problems might fall in this category vs not.
The construction details of the datasets is omitted from the main paper and is delegated to the appendix. At least a brief description is expected in the main paper.
Overall, I think I am concerned about the limited scope of the benchmarks.

**Questions:**

I would encourage the authors to consider a larger scope beyond 1-sufficient CSPs or clearly articulate why this is a significant enough problem space.

---

> ### Author Response · Authors · 2024-11-26
>
> > limiting the scope to 1-sufficient CSPs… concerned about the limited scope of the benchmarks
>
> Please see the general reply “Scope limited to 1-sufficient CSPs”.
>
>
> > unclear how much of the practical problem space does this cover
>
> Please see the general reply “applicability to real-world scenarios”.
>
> > Gaps in writing that make it difficult to follow. Add examples for the challenges around identifying necessary information to acquire and the lack of ground truth
>
> Thank you for this suggestion. Please note that we have illustrative examples in Figure 1 and the beginning of section 3 for the challenge of identifying necessary information.
>
> The lack of ground truth is due to the well-known rater disagreement problem for subjective tasks [2,3,4,5,6,7]. For generic information gathering tasks, examples for the challenges around the lack of ground truth include
> 1. The user gives an underspecified query: “give me recommendations for dinner.” ChatGPT currently presents a list of dishes and asks the question “What are you in the mood for?” One person might find the question helpful, but another person might find it too generic and unhelpful.
> 2. The user gives an underspecified query: “plan a trip to Japan.” ChatGPT currently presents a long list of steps, and asks “Would you like a more tailored plan or help booking tickets?” One might find it helpful since they want ChatGPT to book tickets, but another might find this question not eliciting the most important information, like time of travel.
>
> This can happen for many underspecified queries that involve subjectivity in evaluation. We have now included an example in the introduction and cited the above papers.
>
> Please let us know if you have suggestions on other examples to include.
>
> > The notion of constraint satisfaction problems is loosely defined. It is unclear what class of practical problems might fall in this category vs not.
>
> Please see our responses above for “limiting the scope to 1-sufficient CSPs…” and “ unclear how much of the practical problem space does this cover”.
>
>
> > The construction details of the datasets is omitted from the main paper and is delegated to the appendix. At least a brief description is expected in the main paper.
>
> Thanks for this suggestion, we originally moved the construction details to the appendix to save space and not overshadow the main paper with unnecessary details. We preserved what we believe to be the main dataset description in Section 4.
>
> > I would encourage the authors to consider a larger scope beyond 1-sufficient CSPs or clearly articulate why this is a significant enough problem space.
>
> Please see the general reply “Scope limited to 1-sufficient CSPs”.
>
>
> [1] https://en.wikipedia.org/wiki/Constraint_satisfaction_problem
>
> [2] Aroyo, Lora, and Chris Welty. "Truth is a lie: Crowd truth and the seven myths of human annotation." AI Magazine 36.1 (2015): 15-24.
>
> [3] Aroyo, Lora, et al. "Dices dataset: Diversity in conversational ai evaluation for safety." Advances in Neural Information Processing Systems 36 (2024).
>
> [4] Davani, Aida Mostafazadeh, Mark Díaz, and Vinodkumar Prabhakaran. "Dealing with disagreements: Looking beyond the majority vote in subjective annotations." Transactions of the Association for Computational Linguistics 10 (2022): 92-110.
>
> [5] Basile, Valerio, et al. "We need to consider disagreement in evaluation." Proceedings of the 1st workshop on benchmarking: past, present and future. Association for Computational Linguistics, 2021.
>
> [6] Sandri, Marta, et al. "Why don’t you do it right? analysing annotators’ disagreement in subjective tasks." Proceedings of the 17th Conference of the European Chapter of the Association for Computational Linguistics. 2023.
>
> [7] Wan, Ruyuan, Jaehyung Kim, and Dongyeop Kang. "Everyone’s voice matters: Quantifying annotation disagreement using demographic information." Proceedings of the AAAI Conference on Artificial Intelligence. Vol. 37. No. 12. 2023.

---

### Author Response · Authors · 2024-11-26
**General reply**

We thank all reviewers for their insightful feedback and recognizing our key strengths, including

- the importance of evaluating the ability of LLMs to ask clarification questions [Hjhv, shNW, ai7d];
- a novel CSP-based formulation [shNW, Rppr] and benchmark [shNW, Rppr, ai7d];
- analyses and insights on why SOTA LLMs achieve low performance [Hjhv, ai7d];
- well-designed (Rppr), clear and easy to understand (shNW, ai7d​​).

**We believe the ratings are too harsh and not calibrated, and ask the reviewers to please reconsider the ratings.**

Below we include responses to some common points brought up by reviewers.

> Scope limited to 1-sufficient CSPs. [Hjhv, Rppr]

1. reasons for limiting the scope to CSPs:

- A range of reasoning tasks can be formulated as CSPs as shown in our benchmark. Moreover, classic CSPs cover a wide variety of important problems studied in fields like AI and operations research [1].
- As described in Section 3.1, we find that formulating information gathering as CSPs can effectively allow us to disentangle semantic ambiguity and underspecification. We are not aware of other formulations that can separate those two in such a clean way. Please note that this novel formulation is one of our core strengths, highlighted by Reviewer shNW, Rppr.
- As such, we believe that the CSP formulation is a foundational piece and critical tool for studying the information gathering problem in LLMs.

2. reasons for limiting the scope to tasks that only require one question:

- Imagine a task that requires k questions. Once the first question is answered, the task will require k-1 questions. Hence eventually all such tasks reduce to 1-sufficient CSPs.
- Evaluations on 1-sufficient CSPs serve as an upper bound on the performance for k-sufficient CSPs since in order to solve k-sufficient CSPs, one must be able to solve 1-sufficient CSPs. Hence 1-sufficient CSPs would be the first set of problems to tackle to improve the information gathering skills for reasoning problems.
- Human users may find many questions from AI assistants to be annoying. In practice,  AI assistants may ask only a few questions before solving a user-specified task.

We have now clarified these points in the paper (see footnote 1 in introduction) and changed the title of our work to “QuestBench: Can LLMs ask the right question to acquire information in reasoning tasks?” to highlight “the right question” (singular form).

> applicability to real-world scenarios [Hjhv, Rppr]

For this work, we deliberately chose to construct underspecified reasoning tasks, covering practical problems that involve solving grade school math problems and partially observable robot planning (i.e., initial state is not fully known to the robot). These tasks highlight the basic information gathering ability of LLMs, similar to the basic math ability evaluated by GSM8K, the basic logical reasoning skills evaluated by SimpleLogic, etc. To ensure the scope of our work is clearly communicated, we have now changed the title of our work to “QuestBench: Can LLMs ask the right question to acquire information in reasoning tasks?”. Please let us know if you have better suggestions.

The exact practical problem space is difficult to determine without grand scale data collection for what people are using LLMs for. One can argue none of the existing academic LLM benchmarks cover the majority of “practical problems” since they lack the ability to acquire actual human user data in real practical commercial settings.

> How to compute accuracy [shNW, ai7d]

Accuracy is computed by exact match with the ground-truth question. In L257, we explain that “During evaluation, we consider a LLM’s behavior to be correct if they produce a variable in any 1-sufficient set”. To clarify what this means: we prompt LMs to simply pick a variable to ask about, and check if the variable picked by the LM matches a ground-truth sufficient variable. Prompts for each dataset can be found in Appendix B.


> future work on method [shNW, ai7d]

One recommendation is to use LLMs to extract the symbolic CSP for an underspecified task and then run search algorithms to find the right variable to clarify. We have now made this clear in the discussion and conclusion section.

---

### Author Response · Authors · 2024-12-03
**Concerns about the lack of discussion and unfair evaluation**

We appreciate the reviewers' time and effort in providing the initial feedback. We submitted a detailed rebuttal addressing all of the points raised by the reviewers but it is unfortunate that **no reviewer participated in the discussion or acknowledged reading the rebuttal/paper revision**.

While we understand that reviewing for a conference of this scale can be time-consuming and demanding, the absence of any discussion following our rebuttal raises concerns about the fairness and the quality of the evaluation.

Moreover, we worry that our inclusion of formal mathematical formulation prevented some reviewers from understanding the focus and significant contribution of our work, including being the first to rigorously define underspecification in reasoning tasks and reliably evaluate information-gathering abilities of frontier models.

We know it is unlikely, but we hope the AC and reviewers can read the rebuttal and the revised paper, let us know if any additional clarification is needed and fairly recalibrate the ratings. If you finish reading this message, thank you for your attention.

---

> ### Comment · Area_Chair_z5Ej · 2024-12-03
>
> Thanks for calling my attention to this. I'll work with reviewers on it.

---

### Meta-Review · Area_Chair_z5Ej · 2024-12-21

**Metareview:**

(a) Scientific claims and findings:
The paper constructs QUESTBENCH, a benchmark to evaluate LLMs' capability to resolve underspecified tasks through information gathering. It demonstrates that even advanced LLMs perform poorly, especially on computationally intensive tasks, and identifies potential limitations in LLMs' reasoning mechanisms.

(b) Strengths:

- Novel framing of information-gathering as CSPs, distinguishing underspecification from semantic ambiguity.
- Comprehensive benchmark spanning logic, planning, and math reasoning tasks.

(c) Weaknesses:

- Limited scope to 1-sufficient CSPs reduces applicability to real-world problems.
- Absence of natural language tasks, narrowing practical relevance.
- Insufficient exploration of methods to address failures and improve model capabilities.
- Evaluation metrics require further clarification, though addressed during the rebuttal.

(d) Decision: reject
While the benchmark and framing of the problem are valuable contributions, the limited scope and practical relevance, along with the lack of actionable insights for improvement, reduce the paper's impact.

**Additional Comments On Reviewer Discussion:**

Reviewers raised concerns about the limited scope of the benchmark, lack of practical tasks, and absence of insights into improving LLMs' performance. There are also questions about evaluation metrics and failure analysis were also noted.

Authors' responses:

- Clarified the rationale for focusing on 1-sufficient CSPs and its foundational role in solving multi-sufficient CSPs.
- Added examples and explanations for accuracy evaluation.
- Acknowledged limitations in scope but defended the focus on controlled experiments.

Changes made:

- Title was updated to emphasize “the right question” for clarity.
- Additional examples and clarifications were incorporated into the manuscript.

Authors raised concerns about the lack of engagement of reviewers in discussion. Reviewers responded after reminder but decided to keep their scores unchanged.

---

### Decision · Program_Chairs · 2025-01-22

Reject